# RegMixup: Mixup as a Regularizer Can Surprisingly Improve Accuracy & Out-of-Distribution Robustness

**Francesco Pinto**[*]
University of Oxford
francesco.pinto@eng.ox.ac.uk

**Harry Yang**
Meta AI

**Ser-Nam Lim**
Meta AI

**Philip H.S. Torr**
University of Oxford

**Puneet K. Dokania**
University of Oxford & Five AI Ltd.
puneet.dokania@five.ai

## Abstract

*We show that the effectiveness of the well celebrated Mixup [Zhang et al., 2018] can be further improved if instead of using it as the sole learning objective, it is utilized as an additional regularizer to the standard cross-entropy loss. This simple change not only improves accuracy but also significantly improves the quality of the predictive uncertainty estimation of Mixup in most cases under various forms of covariate shifts and out-of-distribution detection experiments. In fact, we observe that Mixup otherwise yields much degraded performance on detecting out-of-distribution samples possibly, as we show empirically, due to its tendency to learn models exhibiting high-entropy throughout; making it difficult to differentiate in-distribution samples from out-of-distribution ones. To show the efficacy of our approach (RegMixup[2]), we provide thorough analyses and experiments on vision datasets (ImageNet & CIFAR-10/100) and compare it with a suite of recent approaches for reliable uncertainty estimation.*

## 1 Introduction

In real-world machine learning applications one is interested in obtaining models that can reliably process novel inputs. However, though deep learning models have enabled breakthroughs in multiple fields, they are known to be unreliable when exposed to samples obtained from a distribution that is different from the training distribution. Usually, the larger the extent of this difference between the train and the test distributions, the more unreliable these models are. This observation has led to a growing interest in developing deep learning models that can provide *reliable* predictions even when exposed to unseen situations [Liu et al., 2020a,b, Wen et al., 2021, Lakshminarayanan et al., 2017]. Most of these approaches either use expensive ensembles, or propose non-trivial modifications to the neural network architectures in order to obtain reliable models. In most cases, they trade in-distribution performance (accuracy) to perform reliably on: (1) out-of-distribution (OOD) samples; and (2) covariate shift (CS) [Quionero-Candela et al., 2009] samples.

Towards developing practically useful and reliable models, we investigate the well known Mixup [Zhang et al., 2018] as it is extremely popular in improving both a model's accuracy and its robustness [Wen et al., 2021, Hendrycks et al., 2020b]. It has already been observed that Mixup can help in retaining good accuracy when the test inputs are affected by superficial variations that do not affect the target label (i.e., they undergo CS) [Hendrycks et al., 2020b]. However, we find

---

[*]FP continued spending time on this project during his internship at Five AI from June 2021 to March 2022.
[2]Code available at: https://github.com/FrancescoPinto/RegMixup

36th Conference on Neural Information Processing Systems (NeurIPS 2022).

that Mixup's reliability degrades significantly when exposed to completely unseen samples with potentially different labels than the ones it was shown during training (OOD). This is undesirable as in such situations we would like our models to reliably reject these inputs instead of making wrong predictions on them. We observe that the primary reason for such poor OOD performance of Mixup is due to its tendency to provide high predictive entropy for almost all the samples it receives. Therefore, it becomes difficult for the model to differentiate in-distribution samples from out-of-distribution ones. We would like to highlight that our observation is in contrast to the prior work [Thulasidasan et al., 2019] which suggests that Mixup provides reliable uncertainty estimates for OOD data as well.

We propose a simple yet effective fix to the aforementioned issue with Mixup: we suggest to train the model on an approximate data-distribution that is an *explicit* mixture of both the empirical and the Mixup vicinal approximations to the data-distribution. We call this approach RegMixup. In a nutshell, it simply combines the Empirical Risk Minimization (ERM) [Vapnik, 1991] and the Vicinal Risk Minimization (VRM) [Chapelle et al., 2000] objectives together. Implementation wise, along with the Mixup objective on the interpolated sample, it requires adding an additional cross-entropy loss on one of the the clean samples. We provide proper justifications behind this proposal and show that such simple modification significantly improves the performance of Mixup on a variety of experiments.

We would like to highlight that one of the core strengths of our approach is its **simplicity**. As opposed to the recently proposed techniques to improve uncertainty estimation like SNGP [Liu et al., 2020a] and DUQ [van Amersfoort et al., 2020], it does not require any modifications to the architecture and is extremely simple to implement. *It also does not trade accuracy in order to improve uncertainty estimates*, and is a single deterministic model, hence, extremely efficient compared to the highly competitive Deep Ensembles (DE) [Lakshminarayanan et al., 2017]. Summary of our contributions:

- We provide a simple modification to Mixup that significantly improves its in-distribution, covariate shift, and out-of-distribution performances.
- Through extensive experiments using ImageNet-1K, CIFAR10/100 and their various CS counterparts along with multiple OOD datasets we show that, overall, RegMixup outperforms recent state-of-the-art single-model approaches. In most cases, it outperforms the extremely competitive and expensive DE as well.

## 2 RegMixup: Mixup as a regularizer

**Preliminary on ERM and VRM**  The principle of risk minimization [Vapnik, 1991] is to estimate a function $f \in \mathcal{F}$ that, for a given loss function $\ell(.,.)$, minimizes the expected risk over the data-distribution $P(\mathbf{x}, \mathbf{y})$. The risk to be optimized is defined as $R(f) = \int \ell(f(\mathbf{x}), \mathbf{y})dP(\mathbf{x}, \mathbf{y})$. Since the distribution $P(\mathbf{x}, \mathbf{y})$ is unknown, a crude yet widely used approximation is to first obtain a training dataset $\mathcal{D} = \{(\mathbf{x}_i, \mathbf{y}_i)\}_{i=1}^n$ sampled from the distribution $P$ and then obtain $f$ by minimizing the *empirical risk* defined as $R_e(f) = 1/n \sum_{i=1}^n \ell(f(\mathbf{x}_i), \mathbf{y}_i)$. This is equivalent to approximating the entire data-distribution space by a finite $n$ number of Delta distributions positioned at each $(\mathbf{x}_i, \mathbf{y}_i)$, written as $P_e(\mathbf{x}, \mathbf{y}) = 1/n \sum_{i=1}^n \delta_{\mathbf{x}_i}(\mathbf{x})\delta_{\mathbf{y}_i}(\mathbf{y})$. This approximation to the risk minimization objective is widely known as the *Empirical Risk Minimization (ERM)* [Vapnik, 1991].

ERM has been successfully used in a plenty of real-world applications and undoubtedly has provided efficient and accurate solutions to many learning problems. However, it is straightforward to notice that the quality of such ERM solutions would rely on how closely $P_e$ mimics the true distribution $P$, and also on the capacity of the function class $\mathcal{F}$. In situations where the function class is extremely rich with high capacity (for example, deep neural networks), and hence prone to undesirable behaviours such as overfitting and memorization, a good approximation to $P$ is generally needed to enforce suitable inductive biases in the model. To this end, for a fixed training dataset, instead of a delta distribution one could potentially fit a richer distribution in the vicinity of each input-output pair to estimate a more informed risk computed in a *region* around each sample. This is precisely the principle behind *Vicinal Risk Minimization (VRM)* [Chapelle et al., 2000]. The approximate distribution in this case can be written as $P_v(\mathbf{x}, \mathbf{y}) = 1/n \sum_{i=1}^n P_{\mathbf{x}_i, \mathbf{y}_i}(\mathbf{x}, \mathbf{y})$, where $P_{\mathbf{x}_i, \mathbf{y}_i}(\mathbf{x}, \mathbf{y})$ denotes the user-defined vicinal distribution around the $i$-th sample[3]. Therefore, the *vicinal risk* boils down to

---

[3]Note, the original VRM paper uses $P_{\mathbf{y}_i}(\mathbf{y}) = \delta_{\mathbf{y}_i}(\mathbf{y})$ which simply is a special case of this notation.

$$R_v(f) = \frac{1}{n} \sum_{i=1}^{n} \int \ell(f(\mathbf{x}), \mathbf{y}) dP_{\mathbf{x}_i, \mathbf{y}_i}(\mathbf{x}, \mathbf{y}). \qquad (1)$$

When the integral in the above summation is intractable, a Monte Carlo estimate with $m$ samples can be used as follows:

$$\int \ell(f(\mathbf{x}), \mathbf{y}) dP_{\mathbf{x}_i, \mathbf{y}_i}(\mathbf{x}, \mathbf{y}) \approx \frac{1}{m} \sum_{j=1}^{m} \ell(f(\bar{\mathbf{x}}_j), \bar{\mathbf{y}}_j); \;\; (\bar{\mathbf{x}}_j, \bar{\mathbf{y}}_j) \sim P_{\mathbf{x}_i, \mathbf{y}_i}(\mathbf{x}, \mathbf{y}). \qquad (2)$$

Several approaches in deep learning can be seen as a special instance of VRM. For example, training a neural network with multiple augmentations is a special case where the augmented inputs are the samples from the unknown vicinal distribution. A widely used application of VRM is the procedure to obtain a *robust base classifier* to design certifiable classifiers.

For example, if $P_{\mathbf{x}_i, \mathbf{y}_i}(\mathbf{x}, \mathbf{y}) = P_{\mathbf{x}_i}(\mathbf{x})\delta_{\mathbf{y}_i}(\mathbf{y})$ and $P_{\mathbf{x}_i}(\mathbf{x})$ is a Gaussian distribution centered at $\mathbf{x}_i$, Eq. (2) can be computed by taking the average loss over gaussian perturbed inputs $\bar{\mathbf{x}}$ while keeping the target labels same. Minimizing such a risk would lead to a classifier that is robust to additive noise bounded within an $\ell_2$ ball. This is exactly the procedure that has been widely adopted in the *randomized smoothing* literature in order to obtain a base neural network for which a *certifiable* smooth classifier can be obtained [Lecuyer et al., 2019, Cohen et al., 2019][4]. Below we discuss another highly effective use case of VRM called *Mixup* which is the main focus of this work.

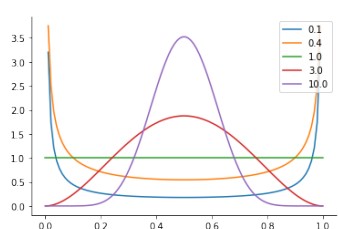

Figure 1: $\texttt{Beta}(\alpha, \alpha)$ pdf for varying $\alpha$.

**Mixup**  Built on the fundamentals of VRM, the vicinal distribution defined in Mixup [Zhang et al., 2018] is as follows:

$$P_{\mathbf{x}_i, \mathbf{y}_i}(\mathbf{x}, \mathbf{y}) = \mathbb{E}_\lambda[(\delta_{\bar{\mathbf{x}}_i}(\mathbf{x}), \delta_{\bar{\mathbf{y}}_i}(\mathbf{y}))],$$

where $\lambda \sim \texttt{Beta}(\alpha, \alpha) \in [0, 1]$, $\alpha \in (0, \infty)$, $\bar{\mathbf{x}}_i = \lambda \mathbf{x}_i + (1 - \lambda)\mathbf{x}_j$ and $\bar{\mathbf{y}}_i = \lambda \mathbf{y}_i + (1 - \lambda)\mathbf{y}_j$. Note that the vicinal distribution here not only depends on $(\mathbf{x}_i, \mathbf{y}_i)$ but also on another input-output pair $(\mathbf{x}_j, \mathbf{y}_j)$ drawn from the same training dataset. For a fixed $\alpha$ (parameter of the Beta distribution, refer Figure 1), implementing Mixup would require taking multiple Monte Carlo samples[5] for each datapoint (refer Eq. (2)) which can be computationally prohibitive. Therefore, in practice, only one sample ($m = 1$) per Beta distribution per pair of samples from a batch is considered at a time. Although this procedure might look like a crude approximation to the original objective, it has resulted in highly promising results in a variety of applications and is very well accepted in the research community. Without undermining the remarkable success of such a simple approach, below we highlight two of its behavioural characteristics that, as we show, might limit its effectiveness:

- **Small cross-validated $\alpha \ll 1$:** The shape of the vicinal distribution depends on the hyperparameter of the Beta distribution, therefore, the values of $\alpha$ decides the strength of the convex interpolation factor $\lambda$. Since high values of $\alpha$ would encourage $\lambda \approx 0.5$ resulting in an interpolated $\bar{\mathbf{x}}$ that is perceptually different from $\mathbf{x}$ (*inducing a mismatch between train and test distributions*), the cross-validated value of $\alpha$ for Mixup in most cases turns out to be very small ($\alpha \approx 0.2$) in order to provide good generalization. Such small values of $\alpha$ leads to sharp peaks at 0 and 1 (refer Figure 1). Therefore, effectively, Mixup ends up *slightly* perturbing a clean sample in the direction of another sample even though the vicinal distribution has the potential to explore a much larger interpolation space.
- **High-entropy behaviour:** As mentioned earlier, $m = 1$ is used in practice, therefore, it is very unlikely that the interpolation factor $\lambda \in \{0, 1\}$ even for small values of $\alpha$. Thus, the model never gets exposed to uninterpolated clean samples during training and hence, it always learns to predict interpolated (or smoothed) labels $\bar{\mathbf{y}}$ for every input. Just like DNNs with cross-entropy loss are overconfident because of their high capacity and target

---

[4]Based on our understanding, the current literature do not mention this procedure as an instance of VRM.
[5]We do not investigate how $m$ depends on $\alpha$.

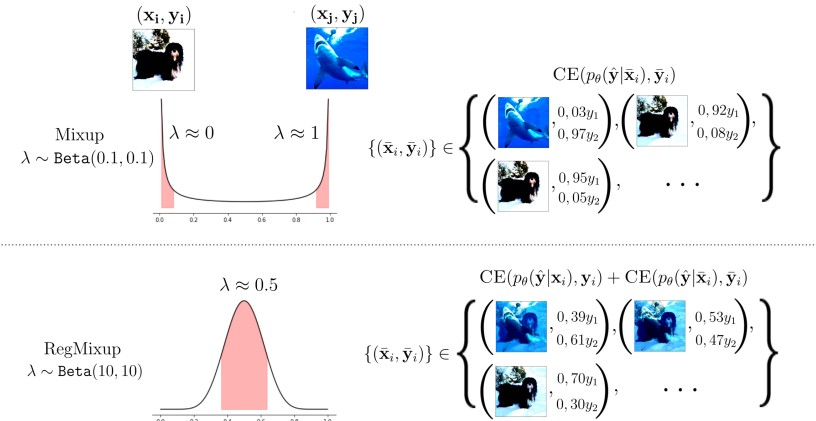

Figure 2: **Mixup vs RegMixup in practice**. Illustration of how the cross-validated $\alpha$ affects the shape of the `Beta` distribution in both cases. Red regions represent 80% of the probability mass. Mixup typically samples $\lambda \approx 0$ or 1, while for RegMixup $\lambda \approx 0.5$. In the first case, one of the two interpolating images dominates the interpolated one; in the latter, a wide variety of images containing features from both the images in the pair are obtained.

Delta distribution [Guo et al., 2017], DNNs with Mixup turns out to be *relatively less confident* because the network retains its high capacity but observes only target smoothed labels. This underconfident behaviour results in high-entropy for both in-distribution and out-of-distribution samples. This is undesirable as it will not allow predictive uncertainty to reliably differentiate in-distribution samples from out-of-distribution ones.

We validate the consequences of the above limitations of Mixup with the following simple experiment. In Figure 3, we provide heat-maps to show how the entropy of the predictive distribution varies when interpolating samples belonging to different classes. The heat-map is created as follows. First, we train a WideResNet28-10 (WRN) [Zagoruyko and Komodakis, 2016] using the CIFAR-10 (C10) dataset. Then, we randomly choose 1K pairs of samples $\{\mathbf{x}_i, \mathbf{x}_j\}$ from the dataset such that $y_i \neq y_j$[6]. For each pair, via convex combination, we synthesize 20 samples $\bar{\mathbf{x}}$s using equally spaced $\lambda$s between 0 to 1. The heat-map is then created using all the 20K samples.

The intensity of each $(\lambda, H)$ bin in the heat-map indicates the number of samples in that bin. Note, DNN (trained with vanilla cross-entropy loss) shows low entropy (thus yielding overconfidence) irrespective of where the interpolated sample lies. However, Mixup shows higher entropy for almost all $\lambda$s (thus yielding lower confidence on in-distribution samples as well). The impact of this behaviour is shown in Table 1. Though Mixup provides improved accuracy compared to DNN for in-distribution (IND) and covariate shift experiments, the high entropy behaviour makes it much worse than DNN when exposed to out-of-distribution detection task. For example, when SVHN is used as the OOD dataset, the performance of Mixup drops by nearly 8.47% compared to DNN. This clearly shows that the distributions of the per-sample entropies on the in-distribution and out-of-distribution sets are harder to separate. However, there is a clear improvement of nearly 5% for covariate shift experiments, implying that Mixup augmentations do improve robustness in this aspect. Note that, in the context of calibration, Mixup's confidence reducing behaviour (equivalent to providing higher predictive entropy) was also observed by [Wen et al., 2021].

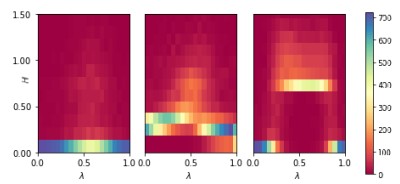

Figure 3: **Heatmaps of the entropy profiles** as the interpolation factor $\lambda \in [0, 1]$ between samples of two classes varies. Left (**DNN**), Middle (**Mixup**), Right (**RegMixup**). Note, RegMixup induces high-entropy barrier separating in-distribution & out-distribution samples.

---

[6]Note, it is highly *likely* that $\mathbf{y}_i \neq \mathbf{y}_j$ even if we do not impose this constraint as the problems under consideration have many classes.

**RegMixup** We now provide a very simple modification to Mixup that not only alleviates its aforementioned limitation on detecting OOD samples, but also significantly improves its performance on in-distribution and covariate-shift samples. We propose to use the following approximation to the data-distribution

$$P(\mathbf{x}, \mathbf{y}) = \frac{1}{n} \sum_{i=1}^{n} (\gamma \delta_{\mathbf{x}_i}(\mathbf{x}) \delta_{\mathbf{y}_i}(\mathbf{y}) + (1 - \gamma) P_{\mathbf{x}_i, \mathbf{y}_i}(\mathbf{x}, \mathbf{y})),$$

where $P_{\mathbf{x}_i, \mathbf{y}_i}(\mathbf{x}, \mathbf{y})$ is the mixup vicinal distribution and $\gamma \in [0, 1]$ is the mixture weight. *The above approximation is simply an explicit assemble of ERM and VRM* based approximations to data-distribution. Though, in theory, VRM subsumes ERM, we argue that because of the crude approximations needed to prevent increasing the training cost, it does not fully realize its potential. Therefore, explicitly combining them might result in a *practically* more expressive model. We provide extensive experimental evidence to support this hypothesis. Implementation wise, for each sample $\mathbf{x}_i$ in a batch, another sample $\mathbf{x}_j$ is drawn randomly from the same batch to obtain interpolated $\bar{\mathbf{x}}_i$ and $\bar{\mathbf{y}}_i$, and then the following loss is minimized

$$\text{CE}(p_\theta(\hat{\mathbf{y}}|\mathbf{x}_i), \mathbf{y}_i) + \eta \, \text{CE}(p_\theta(\hat{\mathbf{y}}|\bar{\mathbf{x}}_i), \bar{\mathbf{y}}_i), \tag{3}$$

where $\text{CE}(.,.)$ denotes the standard cross-entropy loss, the hyperparameter $\eta \in \mathbb{R}_{\geq 0}$, and $p(.)$ the softmax output of a neural network parameterized by $\theta$. Note, dividing Eq. (3) by $(1 + \eta)$ is exactly the same as using a data-distribution with $\gamma = 1/1+\eta$. In practice, we observe the model's performance to not vary much with $\eta$ and using $\eta = 1$ (equivalent to $\gamma = 0.5$) to provide highly effective results. Refer to Algorithm 1 for an overview of the RegMixup training procedure.

*What does this simple modification bring to the model?* (1) VRM now subsumes ERM because the vicinal distribution can perfectly represent the ERM training distribution. This is because of the fact that irrespective of the values of $m$ and $\alpha$, the model will always be exposed to the clean training samples as well. (2) The interpolation factor $\lambda$ can potentially explore a much wider space as the presence of clean samples might help in controlling the performance drop due to the train/test distribution shift. Therefore, if $\alpha \ll 1$ was actually the most effective solution, the cross validation would automatically find it.

*Practical implications of such simple modification on the behaviour and the performance of the model*

- **Large cross-validated $\alpha \gg 1$ :** As anticipated, the model is now able to explore strong interpolations because of the additional cross-entropy term over the unperturbed training data. Interestingly, the cross-validated $\alpha$ that we obtained in fact is very high ($\alpha \in \{10, 20\}$, see Appendix B.3 for cross-validation details) leading to $\lambda \approx 0.5$. Therefore, as opposed to the standard Mixup, RegMixup prefers having *strong diverse interpolations* during training. Refer Figure 2 for visualizations.
- **Maximizing a soft proxy to entropy (knowingly unknowing the unknown):** It is straightforward to notice that a value of $\lambda \approx 0.5$ would lead to $\bar{\mathbf{x}}$ that is a heavy mix of two samples (mimicking OOD samples, refer Fig. 2), and the corresponding target label vector $\bar{\mathbf{y}}$ would have almost equal masses corresponding to the labels of the interpolating samples. Therefore, minimizing $\text{CE}(p_\theta(\mathbf{y}|\bar{\mathbf{x}}_i), \bar{\mathbf{y}}_i)$ in this case is equivalent to *maximizing a soft proxy to entropy* defined over the label support of $\mathbf{y}_i$ and $\mathbf{y}_j$ (note, exact entropy maximization would encourage equal probability masses of 0.5 for these labels). We find this observation intriguing as RegMixup naturally obtains a cross-validated $\alpha$ that leads to a maximum likelihood solution having high-entropy on heavily interpolated samples. This is an extremely desirable property as it allows models to reliably differentiate between in and out distribution samples. Thus, Mixup automatically acts as a **regularizer** in this case.

The entropy heat-map in Figure 3 clearly shows that as opposed to DNN and Mixup, the entropy for RegMixup is very low for $\lambda$ close to either 0 or 1, however, it increases and remains high for all other intermediate interpolation factors, *practically creating an entropy barrier*.

Also, as shown in Table 1, RegMixup provides improvements in both accu-

| Models | C10 (Test) Accuracy (↑) | Cov. Shift C10-C Accuracy (↑) | C100 AUROC (↑) | OOD Detection SVHN AUROC (↑) | T-ImageNet AUROC (↑) |
|---|---|---|---|---|---|
| DNN | 96.14 | 76.60 | 88.61 | 96.00 | 86.44 |
| Mixup | 97.01 | 81.68 | 83.17 | 87.53 | 84.02 |
| RegMixup (Our) | **97.46** | **83.13** | **89.63** | **96.72** | **90.19** |

Table 1: In-distribution, covariate shift, and out-of-distribution detection results for WRN trained on CIFAR-10 (C10). C10-C is the corrupted version of C10.

racy (in-distribution and covariate-shift) and out-of-distribution robustness. An interesting observation is that Mixup's performance on OOD (T-ImageNet) dropped by 2.4% compared to DNN whereas Reg-Mixup performed 3.75% better than DNN. Thus, in this case, it effectively improved the performance of Mixup by nearly 6.15%.

## 3 Related Works

**Covariate-Shift Robustness**   It is well known that neural networks performance can severely degrade when exposed to covariate shift [Taori et al., 2016, Recht et al., 2019, Wang et al., 2019]. In image classification, for instance, covariate shift could occur due to changes in environmental conditions [Hendrycks and Dietterich, 2019] and the image capturing process, data sampling [Recht et al., 2019, 2018, Lu et al.], stylised representations of the objects (e.g. in paintings or drawings) [Wang et al., 2019, Hendrycks et al., 2021] etc. In the field of out-of-distribution generalization, many techniques assume the availability of sets of samples presenting such variations at training time [Shi et al., 2021, Yao et al., 2022, Wang et al., 2021], other techniques suggest leveraging augmentations to mimic a set of variations that would allow the model to learn features that generalize better [Hendrycks et al., 2020b,a] or self-supervision [Hendrycks et al., 2019a, Shen et al., 2022].

**Out-Of-Distribution Detection**   While some authors suggest introducing an OOD reject option [Geifman and El-Yaniv, 2019], the most recent literature tends to focus on leveraging a threshold classifier on an uncertainty measure computed on top of the predictive distribution of a network. For the latter approach, it is therefore convenient for a model to produce high uncertainty predictions when facing OOD samples, and low uncertainty predictions when fed in-distribution samples [Liu et al., 2020a]. Within this family of approaches, the most popular ones are the expensive Deep Ensembles (DE) [Lakshminarayanan et al., 2017]. Note, the computational cost of DE scales linearly with the number of members, and also recent works have observed a few limitations of naively using ensembles directly [Wen et al., 2021, Ashukha et al., 2020]. There are several efficient variants of DE where the ensemble members are obtained either by processing multiple inputs and multiple outputs at the same time [Havasi et al., 2021], adding member-specific components to the network [Wen et al., 2020], or by finding optima in distinct basins of low loss [Huang et al., 2017a]. Significant attention has also been paid to Bayesian techniques where either stochastic gradient descent is being modified to perform approximate inference [Chen et al., 2014, Zhang et al., 2020, Durmus et al., 2016] or an approximation to Gaussian Processes is being utilized (specifically, we focus on SNGP [Liu et al., 2020a]), or Bayesian Logistic Regression is being used during inference via Laplace approximation (specifically, we focus on KFAC-LLLA [Kristiadi et al., 2020]). Techniques utilizing distance functions in the feature space have also been suggested [van Amersfoort et al., 2020].

**Model calibration**   Modern NNs have been shown to be miscalibrated [Guo et al., 2017], i.e. they exhibit a mismatch between the model's confidence and its accuracy. It has been observed that temperature scaling [Guo et al., 2017] or replacing the typical cross-entropy loss [Chung et al., 2021, Thulasidasan et al., 2019, Mukhoti et al., 2020] can be highly effective to reduce this mismatch. Also, ensemble learning has been observed to help in reducing miscalibration [Lakshminarayanan et al., 2017, Wen et al., 2021, Rahaman and Thiery, 2020]. However, it has been shown that good calibration on in-distribution data does not necessarily imply good calibration performance under covariate-shift [Ovadia et al., 2019].

## 4 Experiments

**Datasets and Network Architectures**   We employ the widely used WideResNet28-10 (WRN) [Zagoruyko and Komodakis, 2016] and ResNet50 (RN50) [He et al., 2016] architectures. We train them on CIFAR-10 (C10) and CIFAR-100 (C100) datasets. We employ RN50 to perform experiments on ImageNet-1K [Deng et al., 2009] dataset. We report the average of all the metrics computed on 5 seeds. For further details about the code base and the hyperparameters, refer to Appendix B.

For **Covariate Shift (CS)** experiments on models trained on C10 and C100, we resort to the widely used CIFAR10-C (C10-C) and CIFAR100-C (C100-C) datasets, corrupted versions of C10 and C100 [Hendrycks and Dietterich, 2019]. These datasets are made by applying 15 synthetically generated but *realistic corruptions* at 5 degrees of intensity on the test sets of C10 and C100,

respectively. For CIFAR-10, we also use the CIFAR-10.1 (C10.1) [Recht et al., 2018] and CIFAR-10.2 (C10.2) [Lu et al.] datasets designed to test the generalization of CIFAR-10 classifiers to *natural covariate shifts*. To the best of our knowledge there is no such analogous dataset for CIFAR-100. For ImageNet-1K experiments, we use the widely considered ImageNet-A (A) [Hendrycks et al., 2019b], ImageNet-R (R) [Hendrycks et al., 2021], ImageNetv2 (V2) [Recht et al., 2019], and ImageNet-Sketch (SK) [Wang et al., 2019] datasets for covariate shift experiments. We report the calibration experiments on both in-distribution and covariate shifted inputs in Appendix A.

For **Out-of-Distribution (OOD)** detection, following SNGP [Liu et al., 2020a], we use C100 and SVHN Netzer et al. [2011] as OOD datasets for models trained on C10. Similarly, for models trained on C100, we use C10 and SVHN as OOD datasets. Additionally, we also consider the Tiny-ImageNet (T-ImageNet) dataset [Le and Yang, 2015] as OOD set in both the cases. For models trained on ImageNet-1K, we use ImageNet-O (O) [Hendrycks et al., 2019b] as the OOD dataset.

**Methods considered for comparisons**   Besides the natural comparison of our method with Mixup [Zhang et al., 2018] and networks trained via ERM on the standard cross-entropy loss (which we will refer to as DNN), we consider several other methods from the OOD detection and CS literature. For models trained on C10 and C100, we consider:

- DNN-SN and DNN-SRN: taking inspiration from [Liu et al., 2020a], we consider DNN models trained with Spectral Normalization (SN) [Miyato et al., 2018a] and Stable Rank Normalization (SRN) [Sanyal et al., 2020] to control the bi-Lipschitz constant of the networks, which has been shown to affect the generalization properties of neural networks.
- SNGP: Spectrally Normalized Gaussian Process  [Liu et al., 2020a].
- DUQ: Deterministic Uncertainty Quantification [van Amersfoort et al., 2020].
- KFAC-LLLA: KFAC-Laplace Last Layer Approximation [Kristiadi et al., 2020] makes a model Bayesian at test time by taking the Laplace approximation of the last layer [Ritter et al., 2018]. We provide a simple outline of this approach in Appendix E.
- AugMix [Hendrycks et al., 2020b]: A data augmentation technique that applies randomized augmentations to an input while enforcing them to be consistent during training. With the recommended hyperparameters in the paper, it is almost $4\times$ slower than DNN during training while having the same inference requirements.
- DE: Deep Ensembles [Lakshminarayanan et al., 2017] with 5 members, requiring $5\times$ more compute than most single-model approaches such as DNN.

We would like to mention that, compared to vanilla DNN, our approach (RegMixup) is almost $1.5\times$ slower and Mixup is about $1.2\times$ slower during training, while having the same inference requirements. Due to the high compute requirements, for ImageNet-1K we consider DNN, Mixup and the two other strongest baselines: AugMix and DE. We also cross-validate the hyperparameters on a $10\%$ split of the test set, which is removed at test time.

**Few missing experiments:**   Below we provide extensive experiments for proper benchmarking. The number of datasets and architectures we use lead to many combinations, a few of which we were not able to produce good results for (even after extensive hyperparameter search), hence, some of the entries in the tables are missing. For example, we could not make DUQ work on C100 as it exhibited unstable behaviour. We could not produce promising results for SNGP on RN50 CIFAR experiments using their official implementation. Similarly for AugMix RN50 experiments on C10 and C100. We chose not to report these suboptimal numbers. Further details can be found in Appendix B.

**Table entries:**   Bold represents the best among all the single-model approaches, and underlined represents the best among all including the expensive Deep Ensembles.

Note, we do not consider methodologies requiring access to an external dataset during training (either for CS or OOD ) as not only this would be an unfair comparison[7], but we believe assuming such knowledge is against the goal of this work, which is to develop models robust to unknown scenarios.

---

[7]since all the methods we consider do not leverage this information. RegMixup only relies on in-distribution training data, just like other approaches, and makes no assumption on the type of CS nor on the OOD inputs.

### 4.1 RegMixup Improves Accuracy on In-distribution and Covariate-shift Samples

**Small-scale (CIFAR) Experiments on In-distribution Data** In Table 2, we compare the accuracy of various approaches on the in-distribution test sets of C10 and C100, respectively.

Clearly,

- RegMixup outperforms Mixup in all these experiments. In fact, RegMixup is the best performing one among all the single-model approaches.
- These improvements are non-trivial. For instance, on WRN trained on C100, it outperforms DNN and Mixup by 1.67% and 0.65%, respectively. It outperforms SNGP with a significant margin of 4.05%.

|  | IND Accuracy | | | |
|---|---|---|---|---|
|  | WRN28-10 | | RN50 | |
| Methods | C10 (Test) | C100 (Test) | C10 (Test) | C100 (Test) |
|  | Accuracy (↑) | | Accuracy (↑) | |
| DNN | 96.14 | 81.58 | 95.19 | 79.19 |
| Mixup | 97.01 | 82.60 | 96.05 | 80.12 |
| RegMixup (our) | **97.46** | **83.25** | **96.71** | **81.52** |
| DNN-SN | 96.22 | 81.60 | 95.20 | 79.27 |
| DNN-SRN | 96.22 | 81.38 | 95.39 | 78.96 |
| SNGP | 95.98 | 79.20 | - | - |
| DUQ | 94.7 | - | - | - |
| KFAC-LLLA | 96.11 | 81.56 | 95.21 | 79.41 |
| Augmix | 96.40 | 81.10 | - | - |
| DE (5×) | 96.75 | 83.85 | 96.23 | 82.09 |

Table 2: Accuracies (%) on IND samples for models trained on C10 and C100

Note how the single-model approaches specifically designed to provide reliable predictive uncertainty estimations (for example, SNGP, DUQ, KFAC-LLLA) underperform even compared to the vanilla DNN in terms of in-distribution accuracy. In order to provide improved uncertainty estimates (as we will soon show), they trade clean data accuracy. This type of behaviour is not observed in RegMixup.

**Small-scale (CIFAR) Covariate Shift Experiments** For C10-C and C100-C, as typical in the literature, we report the accuracy averaged over all the corruptions and degrees of intensities in Table 3. It is evident that our approach produces a remarkable improvement in the average accuracy compared to **all** the baselines (except AugMix, we discuss later why that might be the case). For instance, for C100-C WRN experiments, our method achieves an accuracy improvement of **6.9%** over DNN, of **3.86%** over DE, and of **2.45%** over Mixup. Similarly, for C10-C WRN, our method achieves an improvement of almost **6.53%** over DNN, of **4.81%** over DE, and of **1.45%** over Mixup.

For natural covariate shift datasets C10.1 and C10.2 as well, RegMixup outperforms **all** the baselines (including AugMix). For instance, on C10.2, it obtains an improvement of **3.26%** over DNN, of **1.5%** over Mixup, and of **2.46%** over the expensive DE.

*Why AugMix performs extraordinarily well on synthetically corrupted C10-C and C100-C but not on natural distribution shift C10.1 and C10.2?* Looking at Table 3 one can observe Aug-Mix's extremely good performance on the C10-C and C100-C. However, at the same time, the model is underperforming with respect to Reg-Mixup on C10.1 and C10.2. Similarly, AugMix is outperformed by RegMixup on all ImageNet CS experiments (and OOD as will be shown soon). This seems to suggest that although the augmentations used during the training of Aug-Mix are not exactly same as that of the corrupted test dataset, they tend to benefit from synthetic forms of covariate shifts, hence the dramatic improvement in these particular scenarios.

|  | Covariate Shift Accuracy | | | | | | | |
|---|---|---|---|---|---|---|---|---|
|  | WRN28-10 | | | | R50 | | | |
| Methods | C10-C | C10.1 | C10.2 | C100-C | C10-C | C10.1 | C10.2 | C100-C |
|  | Accuracy (↑) | | | | Accuracy (↑) | | | |
| DNN | 76.60 | 90.73 | 84.79 | 52.54 | 75.18 | 88.58 | 83.31 | 50.62 |
| Mixup | 81.68 | 91.29 | 86.55 | 56.99 | 78.63 | 90.03 | 84.61 | 53.96 |
| RegMixup (our) | 83.13 | **92.79** | **88.05** | 59.44 | **81.18** | **91.58** | **86.72** | **57.64** |
| DNN-SN | 76.56 | 91.01 | 84.72 | 52.61 | 74.88 | 88.26 | 82.96 | 50.55 |
| DNN-SRN | 77.21 | 90.88 | 85.24 | 52.54 | 75.40 | 88.61 | 83.49 | 50.48 |
| SNGP | 78.37 | 90.80 | 84.95 | 57.23 | - | - | - | - |
| DUQ | 71.6 | - | - | 50.4 | - | - | - | - |
| KFAC-LLLA | 76.56 | 90.68 | 84.68 | 52.57 | 75.18 | 88.34 | 83.50 | 50.85 |
| AugMix | **90.02** | 91.6 | 85.9 | **68.15** | - | - | - | - |
| DE (×5) | 78.32 | 92.17 | 85.59 | 55.58 | 77.63 | 90.05 | 85.00 | 53.91 |

Table 3: Accuracies (%) on covariate shifted samples for models trained on C10 and C100.

|  | IND Acc. | Covariate Shift Acc | | | | OOD Det. |
|---|---|---|---|---|---|---|
|  | ImageNet-1K | R | A | V2 | Sk | O |
|  | Acc (↑) | Acc (↑) | Acc (↑) | Acc (↑) | Acc (↑) | AUROC (↑) |
| DNN | 76.60 | 36.41 | 2.76 | 64.53 | 24.72 | 55.97 |
| Mixup | 77.15 | 39.05 | 3.29 | 64.58 | 26.34 | 55.54 |
| RegMixup (our) | **77.68** | **39.76** | **5.96** | 65.66 | 26.98 | **57.05** |
| AugMix | 76.88 | 38.29 | 2.63 | 64.94 | 25.61 | 56.91 |
| DE (5×) | 78.22 | 38.94 | 2.11 | 66.68 | 27.03 | 53.29 |

Table 4: ImageNet accuracies (%) on IND and CS samples, and OOD detection performance.

**Large-scale (ImageNet-1K) Experiments on In-distribution Data** As shown in Table 4, RegMixup scales to ImageNet-1K and exhibits improved accuracy with respect to both Mixup, DNN, and AugMix. In particular, it is **1.08%** better than DNN, **0.53%** better than Mixup and **0.80%** better than AugMix. DE in this case is the best performing one.

**Large-scale (ImageNet-1K) Covariate Shift Experiments** In Table 4 we report the results for common ImageNet-1K covariate-shift test sets. As it can be seen, RegMixup is either the best performing one among all the single-model approaches, or it is the absolute winner including DE.

| Out-of-Distribution | WRN28-10 | | | | | | RN50 | | | | | |
|---|---|---|---|---|---|---|---|---|---|---|---|---|
| | CIFAR10 (In-Distribution) | | | CIFAR100 (In-Distribution) | | | CIFAR10 (In-Distribution) | | | CIFAR100 (In-Distribution) | | |
| | C100 | SVHN | T-ImageNet | C10 | SVHN | T-ImageNet | C100 | SVHN | T-ImageNet | C10 | SVHN | T-ImageNet |
| **Methods** | AUROC (↑) | | | AUROC (↑) | | | AUROC (↑) | | | AUROC (↑) | | |
| DNN | 88.61 | 96.00 | 86.44 | 81.06 | 79.68 | 80.99 | 88.61 | 93.20 | 87.82 | 79.33 | 82.45 | 79.89 |
| Mixup | 83.17 | 87.53 | 84.02 | 78.37 | 78.68 | 80.61 | 84.24 | 89.40 | 84.89 | 77.02 | 76.86 | 80.14 |
| RegMixup (**our**) | 89.63 | **96.72** | 90.19 | 81.27 | 89.32 | 83.13 | 89.63 | 95.39 | 90.04 | 79.44 | 88.66 | 82.56 |
| DNN-SN | 88.56 | 95.59 | 87.71 | 81.10 | 83.43 | 82.26 | 88.19 | 93.46 | 87.55 | 79.20 | 80.78 | 79.90 |
| DNN-SRN | 88.46 | 96.12 | 87.43 | 81.26 | 85.51 | 82.41 | 88.82 | 93.54 | 87.82 | 78.77 | 82.39 | 79.70 |
| SNGP | **90.61** | 95.25 | 90.01 | 79.05 | 86.78 | 82.60 | - | - | - | - | - | - |
| KFAC-LLLA | 89.33 | 94.17 | 87.81 | 81.04 | 80.32 | 81.57 | 89.54 | 93.13 | 88.32 | 79.30 | 82.80 | 80.17 |
| Aug-Mix | 89.78 | 91.3 | 88.99 | 81.10 | 76.64 | 80.56 | - | - | - | - | - | - |
| DE (5×) | 91.25 | 97.53 | 89.52 | 83.26 | 85.07 | 83.40 | 91.38 | 96.90 | 90.5 | 81.93 | 85.08 | 82.15 |

Table 5: Out-of-distribution detection results (%) for WideResNet28-10 and ResNet50 for models trained on C10 and C100. See Appendix B for the cross-validated hyperparameters.

For instance, on ImageNet-A, RegMixup performs **2.67%** better than Mixup and **3.20%** better than DNN. Similarly, on ImageNet-V2 it performs **1.08%** better than Mixup and **1.13%** more than DNN. RegMixup also outperforms AugMix on all the considered datasets, while is outperformed by DE on ImageNet-V2 (by **1.02%**) and performs competitively on ImageNet-SK.

*These experiments clearly show the strong generalization of RegMixup under various in-distribution and CS experiments. They also show that it does not trade clean data accuracy to do so.*

## 4.2 Out-of-Distribution Detection Experiments

Following the standard procedure [Liu et al., 2020a], we report the performance in terms of AUROC[8] for the binary classification problem between in- and out-distribution samples. The predictive uncertainty of the model is typically used to obtain these curves. Given an uncertainty measure (normally entropy, refer Appendix C for an extensive discussion), it is important for models to be more uncertain on OOD samples than on in-distribution samples to be able to distinguish them accurately. This behaviour would lead to a better AUROC.

We report the OOD detection results for small-scale experiments (CIFAR 10/100) in Table 5, and for large-scale experiments (ImageNet) in Table 4. Compared to the single-model approaches, it is clear that RegMixup outperforms all the baselines in both small-scale and large-scale experiments, except only in one situation (C100 as OOD, trained on WRN) where SNGP outperformed it by an AUROC of $0.98$. In fact, on ImageNet experiments, RegMixup outperformed all the baselines including DE. We would also like to highlight that RegMixup provides significantly better performance compared to Mixup on all these experiments. For example, the improvement is more than $9\%$ when SVHN is treated as the OOD dataset for WRN trained on either C10 or C100 (refer Table 5). Similarly, we can observe dramatic improvements on other experiments as well.

## 5 Conclusive Remarks

We proposed RegMixup, an extremely simple approach that combines Mixup with the standard cross-entropy loss. We conducted a wide range of experiments and showed that RegMixup significantly improved the reliability of uncertainty estimates of deep neural networks, while also provided a notable boost in the accuracy. We showed that RegMixup did not just outperform Mixup, it also outperformed most recent state-of-the-art approaches in providing reliable uncertainty estimates.

We hope that our work opens possibilities to explore situations where ERM and VRM can explicitly be combined together for practical benefits. An example would be label smoothing [Müller et al., 2019] that can be seen as an instance of VRM where the vicinal distribution is over the labels and the marginal distribution of the input samples (e.g., images) is approximated using deltas. In Appendix H we conduct a preliminary analysis using CutMix Yun et al. [2019] and recent techniques to train Vision Transformers Dosovitskiy et al. [2021] that alternate between Mixup and CutMix. Another possible future direction would be to use bi-modal vicinal distribution along with importance sampling in order to ensure that the unperturbed samples as well are used during training with high probability. The observation that RegMixup ends up acting as a soft proxy to entropy maximizer for interpolated samples, a potential extension of our work regards the possibility of exploring different interpolation techniques to enforce high entropy behaviour on those regions of the input space.

---

[8]Area Under Receiver Operating Characteristic curve.

## Acknowledgements

This work is supported by the UKRI grant: Turing AI Fellowship EP/W002981/1 and EPSRC/MURI grant: EP/N019474/1. We would like to thank the Royal Academy of Engineering, FiveAI and Meta AI. Francesco Pinto's PhD is funded by the European Space Agency (ESA). Meta AI authors are neither supported by the UKRI grants nor have any relationships whatsoever to the grant. Thanks to John Redford, Jon Sadeghi, Kemal Oksuz, Zygmunt Lenyk, Guillermo Ortiz-Jimenez, Amartya Sanyal, Pau de Jorge, and David Lopez-Paz for their valuable comments on the paper.

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
