# A  Calibration on In-distribution and Covariate-shift datasets

Additionally, we provide the calibration performance of various competitive approaches. Briefly, calibration quantifies how similar a model's confidence and its accuracy are [Osborne, 1991]). To measure it, we employ the recently proposed Adaptive ECE (AdaECE) [Mukhoti et al., 2020]. For all the methods, the AdaECE is computed after performing temperature scaling [Guo et al., 2017] with a cross-validated temperature parameter. We also provide the AdaECE without temperature scaling in Appendix D. For completeness, we also report the ECE in Appendix G.

In terms of calibration on in-domain test sets (refer Tables 6 and 7), our method either remarkably improves the AdaECE with respect to Mixup and DNN, or performs competitively (on ImageNet-1K).

Under covariate shift (refer Tables 7 and 8), on corrupted inputs, RegMixup underperforms with respect to Mixup on C10-C, but not on C100-C. On all other C10 covariate shift datasets, RegMixup outperforms both Mixup and DNN. Considering also the other baselines, except for the case of C10-C (in which AugMix significantly outperforms any other baseline on WRN28-10), our method provides the best calibration in all

| | IND | | | |
|---|---|---|---|---|
| | **WRN28-10** | | **RN50** | |
| | C10 (Test) | C100 (Test) | C10 (Test) | C100 (Test) |
| Methods | AdaECE (↓) | | | |
| DNN | 1.34 | 3.84 | 1.45 | 2.94 |
| Mixup | 1.16 | 1.98 | 2.17 | 7.47 |
| RegMixup (our) | **0.50** | **1.76** | **0.94** | **1.53** |
| SNGP | 0.87 | 1.94 | - | - |
| Augmix | 1.67 | 5.54 | - | - |
| DE (5×) | 1.04 | 3.29 | 1.28 | 2.98 |

Table 6: CIFAR IND calibration performance (%).

| | IND | Covariate Shift | | | |
|---|---|---|---|---|---|
| | ImageNet-1K (Test) | ImageNet-R | ImageNet-A | ImageNet-V2 | ImageNet-Sk |
| | AdaECE (↓) | AdaECE (↓) | AdaECE (↓) | AdaECE (↓) | AdaECE (↓) |
| DNN | 1.81 | 13.56 | 44.90 | 4.13 | 14.48 |
| Mixup | **1.29** | 12.08 | 44.63 | 4.28 | 15.26 |
| RegMixup (our) | 1.37 | 13.30 | **41.18** | **3.38** | 15.35 |
| AugMix | 2.05 | **11.24** | 42.83 | 3.94 | **14.26** |
| DE (5×) | 1.38 | 13.55 | 42.88 | 4.02 | 17.32 |

Table 7: ImageNet calibration performance on IND and CS datasets (%).

| | Covariate Shift | | | | | | | |
|---|---|---|---|---|---|---|---|---|
| | **WRN28-10** | | | | **R50** | | | |
| | C10-C | C10.1 | C10.2 | C100-C | C10-C | C10.1 | C10.2 | C100-C |
| Methods | AdaECE (↓) | | | | AdaECE (↓) | | | |
| DNN | 12.62 | 4.13 | 8.81 | 9.94 | 12.29 | 4.36 | 8.89 | 19.76 |
| Mixup | 7.93 | 4.39 | 7.44 | 10.45 | **10.75** | 5.72 | 10.59 | 12.63 |
| RegMixup (our) | 9.08 | **2.57** | **6.83** | **7.93** | 11.37 | **2.89** | **6.74** | **11.47** |
| SNGP | 11.34 | 4.36 | 8.33 | 10.43 | - | - | - | - |
| AugMix | **4.56** | 3.23 | 8.33 | 12.15 | - | - | - | - |
| DE (5 ×) | 10.31 | 2.60 | 7.50 | 12.36 | 12.68 | 4.10 | 6.94 | 12.36 |

Table 8: CIFAR CS calibration performance (%).

other cases. For example, on C100-C experiments on WRN28-10, in terms of AdaECE, RegMixup obtains a **4.43%** improvement over DE, **2.52%** over Mixup, and **2.47%** over SNGP. Though RegMixup outperformed all other approaches in 12 scenarios out of total 17 presented here, it is clear that there is no single method that outperforms any other in all the considered settings.

# B  Experimental Details

## B.1  Code-base

The **RegMixup** training procedure is outlined in Algorithm 1.

For fair comparisons, when training on C10 and C100, we developed our own code base for all the approaches (except SNGP, DUQ and AugMix) and performed an extensive hyperparameter search to obtain the strongest possible baselines.

We would like to highlight that it was not easy to make a few recent state-of-the-art approaches work in situations different from the ones they reported in their papers as these approaches mostly required non-trivial changes to the architectures and additional sensitive hyperparametes. We also observed that their performances did not easily translate to new situations. Below we highlight few of these issues we faced and the measures we took for comparisons.

**For DUQ**, the original paper did not perform large scale experiments similar to ours. Unfortunately, we could not manage to make their code work on C100 as the training procedure seemed to be *unstable*. For this reason, *wherever possible*, we borrowed the numbers for DUQ from the SNGP paper. Please note that the authors of SNGP performed non-trivial modifications to the original DUQ methodology to make it work on C100.

**For SNGP**, we used the publicly available code following exactly the same procedure as mentioned in their original paper. The code *diverges slightly* from the procedure described in their paper, hence the slight differences in the performance. The only modification we performed to the official code-base was to make the inference procedure consistent with the one described in the paper: indeed, in their

**Algorithm 1** RegMixup training procedure

---

**input** Batch $\mathcal{B}$, $\alpha$, $\theta_t$
 $\bar{\mathcal{B}} \leftarrow \emptyset$
 $\lambda_0 \sim \texttt{Beta}(\alpha, \alpha)$
 **for** $\forall(\mathbf{x}_i, \mathbf{y}_i) \in \mathcal{B}$ **do**
  randomly select $(\mathbf{x}_j, \mathbf{y}_j)$ from $\mathcal{B} \backslash (\mathbf{x}_i, \mathbf{y}_i)$
  $\bar{\mathcal{B}} \leftarrow \bar{\mathcal{B}} \cup (\lambda_0 \mathbf{x}_i + (1 - \lambda_0)\mathbf{x}_j, \lambda_0 \mathbf{y}_i + (1 - \lambda_0)\mathbf{y}_j)$
 **end for**
 $\mathcal{L} = \text{CE}(\mathcal{B}) + \text{CE}(\bar{\mathcal{B}})$ *// Loosely speaking, compute cross-entropy loss on both the batches*
 return $\theta_{t+1}$ obtained by updating $\theta_t$ by optimizing the above loss

---

code they implement a mean-field approximation to estimate the predictive distribution [Lu et al., 2020], while in their paper they use Monte Carlo Integration with a number of samples equal to the number of members in the ensembles they use as a baseline, which provides better calibration. The rationale is that we could not find an obvious way to tune the mean-field approximation hyperparameters to improve at the same time both the calibration and OOD detection performance (indeed, *the mean-field approximation imposes a trade-off between calibration and OOD detection performance*). Additionally, since the standard KFAC-LLLA uses the same Monte Carlo Integration procedure, we opted for the latter for a fair comparison. For the SNGP RN50 experiments, we tried running the official implementation on C10 and C100, but we could not make SNGP converge to SOTA accuracy values. The authors of SNGP did not provide experiment results on C10 and C100 on RN50. Hence we decided *not to report* these experiments for SNGP.

**For the KFAC-LLLA** we leverage the official repository[9] [Hobbhahn et al., 2021] and the Backpack library [Dangel et al., 2020] for the computation of the Kronecker-Factored Hessian.

**For AugMix**, we used their code base and the exact training procedure. AugMix seems to be sensitive to hyperparameters of the training procedure as we could could not get the considered architectures to converge to acceptable accuracy levels under the training regime we used for **all** other baselines. Even with the recipes provided in the AugMix paper, we could not get it to converge to competitive accuracy levels when using RN50 on C10 and C100 hence we decided *not to report* these experiments for AugMix.

## B.2 Optimization

For **C10 and C100** training, we use SGD with Nesterov momentum 0.9 for 350 epochs and a weight decay of $5 \times 10^{-4}$. For WRN, we apply a dropout $p = 0.1$ at train time. For all our experiments we set the batch size to $128^{10}$. At training time, we apply standard augmentations `random crop` and `horizontal flip` similar to [Liu et al., 2020a]). The data is appropriately normalized before being fed to the network both at train and test time.

For **ImageNet-1K** training, we use SGD with momentum for 100 epochs, learning rate 0.1, cosine learning scheduler, weight decay of $1 \times 10^{-4}$, batch size 128 and image size $224 \times 224$. We use `color jitter`, `random horizontal flip` and `random crop` for augmentation. We leverage the `timm` library for training [Wightman, 2019] all the considered methods with Automatic Mixed Precision to accelerate the training.

## B.3 Hyperparameters

- For DNN-SN and DNN-SRN the spectral norm clamping factor (maximum spectral norm of each linear mapping) $c \in \{0.5, 0.75, 1.0\}$ and the target of stable rank $r \in \{0.3, 0.5, 0.7, 0.9\}$ (as $r = 1$ for SRN is the same as applying SN with $c = 1.0$). Refer to Miyato et al. [2018b] and [Sanyal et al., 2020] for details about these hyperparameters.
- For Mixup, we consider a wide range of Beta distribution hyperparameter $\alpha \in \{0.1, 0.2, 0.3, 0.4, 0.5, 1, 5, 10, 20\}$.

---

[9]`https://github.com/19219181113/LB_for_BNNs`
[10]For SNGP and DUQ, we use the hyperparameters suggested in their original papers.

| Train Data/ Architecture | Hyperparas | C10 | | C100 | | ImageNet |
|---|---|---|---|---|---|---|
| | | WRN | R50 | WRN | R50 | |
| DNN | $T$ | 1.32 | 1.51 | 1.33 | 1.42 | 1.19 |
| DNN-SN | $c$ | 0.5 | 0.5 | 0.5 | 0.5 | - |
| | $T$ | 1.42 | 1.51 | 1.21 | 1.42 | - |
| DNN-SR | $r$ | 0.3 | 0.3 | 0.3 | 0.3 | - |
| | $T$ | 1.33 | 1.41 | 1.22 | 1.42 | - |
| DE | $T$ | 1.31 | 1.42 | 1.11 | 1.21 | 1.29 |
| SNGP | $T$ | 1.41 | - | 1.52 | - | - |
| Mixup | $\alpha$ | 0.3 | 0.3 | 0.3 | 0.3 | 0.1 |
| | $T$ | 0.73 | 0.82 | 1.09 | 1.21 | 1.06 |
| RegMixup | $\eta = 1, \alpha$ | 20 | 20 | 10 | 10 | 10 |
| | $T$ | 1.12 | 1.31 | 1.23 | 1.21 | 1.14 |
| KFAC-LLLA | #samples | 1000 | 1000 | 1000 | 1000 | |
| | $\sigma_0$ | 1 | 0.6 | 4 | 0.1 | - |

Table 9: Cross-validated hyperparameters. Note, $T$ and $\sigma_0$ are cross-validated by minimizing the ECE. All other hyperparameters have been tuned to maximise the accuracy.

- For RegMixup we consider the Beta distribution hyperparameters to be $\alpha \in \{0.1, 0.2, 0.3, 0.4, 0.5, 1, 5, 10, 15, 20, 30\}$, and the mixing weight $\eta \in \{0.1, 1, 2\}$.
- For KFAC-LLLA we take 1000 samples from the distribution. Although the number might seem quite high, we could not notice significant improvements using a lower number of samples. We tuned the prior variance $\sigma_0$ needed for the computation of the Laplace approximation minimising the ECE on the validation set. We also tried using the theoretical value $\sigma_0 = 1/\tau$ [Kristiadi et al., 2020], where $\tau$ represents the weight decay, but it produced inferior results with respect to our cross-validation procedure. We provide an overview of the KFAC-LLLA in Appendix E.
- For Deep Ensembles we use 5 members.
- When temperature scaling is applied, the temperature $T$ is tuned on the validation set, minimising the ECE (we considered values ranging from 0.1 to 10, with a step size of 0.001). For Deep Ensembles, we first compute the mean of the logits, then scale it by the temperature parameter before passing it through the softmax.

All the cross-validated hyperparameters are reported in Table 9. The cross-validation is performed with stratified-sampling on a 90/10 split of the training set to maximise accuracy[11] on C10 and C100. For ImageNet, we split the test set using the same proportion to obtain the validation set, which is then removed from the test set during evaluation. It is important to observe that:

- Cross-validating hyperparameters based solely on the ECE can prefer models with lower accuracy but better calibration. However, a method improving calibration should avoid degrading accuracy.
- Hyperparameters should not be cross-validated based on CS experiments and OOD detection metrics as they these datasets should be unknown during the training and hyperparameter selection procedure as well.

The results of the cross-validation for RegMixup for CIFAR-10, CIFAR-100 can be found in Table 14. To support our claim that Mixup tends to prefer lower $\alpha$ values, we do not only report the cross-validation accuracy on Mixup for the experiments in the main paper (Table 12) but we also report them for two other popular architectures on CIFAR-10 and CIFAR-100: DenseNet-121 Huang et al. [2017b] and PyramidNet200 Han et al. [2017] (Table 13).

## C   Existing Uncertainty Measures

There are various uncertainty measures and there is no clear understanding on which one would be more reliable. In our experiments we considered the following metrics and chose the one best

---

[11]Except for the $\sigma_0$ of the KFAC-LLLA, as we could not observe significant differences in Accuracy between hyperparameters optimising the accuracy and ECE

| Methods | Clean AdaECE ($\downarrow$) | CIFAR-10-C AdaECE ($\downarrow$) | CIFAR 10.1 AdaECE ($\downarrow$) | CIFAR 10.2 AdaECE ($\downarrow$) | | Methods | Clean AdaECE ($\downarrow$) | CIFAR-100-C AdaECE ($\downarrow$) |
|---|---|---|---|---|---|---|---|---|
| **C10 R50** | | | | | | **C100 R50** | | |
| DNN | 3.02 | 17.30 | 7.39 | 12.24 | | DNN | 9.47 | 25.17 |
| Mixup | 2.87 | **11.35** | **4.05** | **7.72** | | Mixup | 7.47 | 21.52 |
| RegMixup (**Ours**) | 1.40 | 11.51 | 4.15 | 8.23 | | RegMixup (**Ours**) | 3.92 | 13.68 |
| DE (5×) | 2.10 | 13.99 | 6.23 | 10.33 | | DE (5×) | 6.50 | 19.76 |
| **C10 WRN** | | | | | | **C100 WRN** | | |
| DNN | 2.27 | 15.92 | 6.00 | 11.00 | | DNN | 5.30 | 17.38 |
| Mixup | 2.23 | **7.93** | 7.22 | **6.58** | | Mixup | 3.60 | 16.54 |
| RegMixup (**Ours**) | **0.67** | 8.36 | **3.02** | 7.03 | | RegMixup (**Ours**) | 2.47 | 10.49 |
| SNGP | 1.51 | 11.33 | 5.59 | 10.85 | | SNGP | 5.65 | 10.89 |
| AugMix | 1.89 | **5.77** | 4.10 | 9.61 | | AugMix | 5.23 | 13.67 |
| DE (5×) | 1.74 | 13.52 | 4.33 | 9.44 | | DE (5×) | 3.92 | 13.47 |

Table 10: CIFAR calibration performance (%) without temperature scaling

suited for each method in order to create the strongest possible baselines. Let $K$ denote the number of classes, $\mathbf{p}_i$ the probability of $i$-th class, and $\mathbf{s}_i$ the `logit` of $i$-th class. Then, these uncertainty measures can be defined as:

- **Entropy**: $H(\mathbf{p}(\mathbf{x})) = -\sum_{i=1}^{K} \mathbf{p}_i \log \mathbf{p}_i$.
- **Dempster-Shafer** [Sensoy et al., 2018]: $\mathtt{DS}(\mathbf{x}) = {}^{K}\!/_{(K+\sum_{i=1}^{K} \exp(\mathbf{s}_i))}$.
- **Energy**: $E(\mathbf{x}) = -\log \sum_{i=1}^{K} \exp(\mathbf{s}_i)$ (ignoring the temperature parameter). This metric was used in [Liu et al., 2020b] for OOD.
- **Maximum Probability Score**: $\mathtt{MPS}(\mathbf{x}) = \max_i \mathbf{p}_i$.
- **Feature Space Density Estimation** (FSDE): Assuming that the features of each class follow a Gaussian distribution, there are several ways one can estimate the *belief* of a test sample belonging to in-distribution data and treat it as a measure of uncertainty. One such approach is to compute the Mahalanobis score $\arg\min_{i \in y}(\phi(\mathbf{x}) - \mu_i)^T \Sigma_i^{-1}(\phi(\mathbf{x}) - \mu_i)$, where $\mu_i$ and $\Sigma_i$ are class-wise mean and the covariance matrices of the *train* data, and $\phi(\mathbf{x})$ is the feature vector.

*In the main paper, we report the OOD detection performance using the* `DS` *score (as it provided slightly improved performance in most cases), except when it damages the performance of a method (e.g. Mixup) or when it does not yield improvements (e.g. KFAC-LLLA). In these situations we use the entropy as the uncertainty measure.*

**Remarks regarding various metrics:** We would like to highlight a few important observations that we made regarding these metrics. **(1)** `DS` **and** $E$ **are equivalent** as they are both decreasing functions of $\sum_{i=1}^{K} \exp(\mathbf{s}_i)$, and since $\log$ does not modify the monotonicity, both will provide the same ordering of a set of samples. Hence, will give the same AUROC values. **(2)** We observed `DS` and $H$ to perform similarly to each other except in a few situations where `DS` provided slightly better results. **(3)** `MPS`, in many situations, was slightly worse. **(4)** We found Gaussian assumption based density estimation to be **unreliable**. Though it provided extremely competitive results for C10 experiments, sometimes slightly better than the `DS` based scores, it performed very poorly on C100. We found this score to be highly unstable as it involves large matrix inversions. We applied the well-known tricks such as perturbing the diagonal elements and the low-rank approximation with high variance-ratio, but the results were sensitive to such stabilization and there is no clear way to cross-validate these hyperparameters.

# D   Calibration Metrics without Temperature Scaling

For completeness, we report the calibration metrics over all the methods and considered datasets without the temperature scaling [Guo et al., 2017] in Tables 10 and 11. Details about the cross-validation procedure used when temperature scaling is applied is provided in Appendix B.

| | **IND** | **Covariate Shift** | | | |
| | **ImageNet-1K (Test)** | **ImageNet-R** | **ImageNet-A** | **ImageNet-V2** | **ImageNet-Sk** |
| | **AdaECE ($\downarrow$)** | **AdaECE ($\downarrow$)** | **AdaECE ($\downarrow$)** | **AdaECE ($\downarrow$)** | **AdaECE ($\downarrow$)** |
| DNN | 4.90 | 20.48 | 52.30 | 9.58 | 22.94 |
| Mixup | **2.28** | **14.70** | 47.41 | 6.46 | **18.26** |
| RegMixup (**our**) | 3.06 | 17.42 | **45.65** | 7.34 | 20.85 |
| AugMix | 4.28 | 19.13 | 51.35 | **3.94** | 21.25 |
| DE (5$\times$) | 3.61 | 17.32 | 51.64 | 7.94 | 19.35 |

Table 11: ImageNet calibration performance (%) without temperature scaling.

# E  Bayesian at Test Time: Last Layer Laplace Approximation

A structural problem of using MLE logistic regression is that the produced uncertainties depend on the decision boundary. On the other hand, replacing the MLE logistic regression with a Bayesian logistic regression and estimating the predictive posterior employing a Laplace approximation allows to produce better uncertainties [Kristiadi et al., 2020]. However, a Bayesian training either requires a modification in the architecture [Liu et al., 2020a] or makes the inference procedure very expensive [Kingma et al., 2015, Gal and Ghahramani, 2016]. Since the objective is to utilize the standard MLE training of neural networks, the idea of Kronecker-Factored Last Layer Laplace Approximation [Kristiadi et al., 2020] is making the network **Bayesian at test time** with almost no additional cost.

Let $\mathbf{w}$ be the parameters of the of the last layer of a neural network, then we seek to obtain the posterior only over $\mathbf{w}$. Let $p(\mathbf{w}|\mathbf{x})$ be the posterior, then the predictive distribution can be written as:

$$p(y = k|\mathbf{x}, \mathcal{D}) = \int \mathtt{softmax}(\mathbf{s}_k)p(\mathbf{w}|\mathcal{D})d\mathbf{w}, \tag{4}$$

where, $\mathbf{s}$ is the logit vector and $\mathtt{softmax}(\mathbf{s}_k)$ is the $k$-th index of the $\mathtt{softmax}$ output of the network.

The Laplace approximation assumes that the posterior $p(\mathbf{s}|\mathcal{D}) \sim \mathcal{N}(\mathbf{s}|\mu, \Sigma)$, where $\mu$ is a mode of the posterior $p(\mathbf{w}|\mathcal{D})$ (found via standard optimization algorithms for NNs) and $\Sigma$ is the inverse of the Hessian $\mathbf{H}^{-1} = -(\nabla^2 \log p(\mathbf{w}|\mathcal{D})|_\mu)^{-1}$. For the formulations and definitions, including the variants with the terms associated to the bias, we refer to [Kristiadi et al., 2020].

For our experiments, we obtain $\Sigma$ using the Kronecker-factored (KF) approximation [Ritter et al., 2018]. Broadly speaking, the KF approximation allows to reduce the computational complexity of computing the Hessian by factorizing the inverse of the Hessian as $\mathbf{H}^{-1} \approx \mathbf{V}^{-1} \otimes \mathbf{U}^{-1}$, then the covariance of the posterior evaluated at a point $\mathbf{x}$ takes following form $\Sigma = (\phi(\mathbf{x})^T \mathbf{V}\phi(\mathbf{x}))\mathbf{U}$. This procedure can be easily implemented using the Backpack library [Dangel et al., 2020] to compute $\mathbf{V}$ and $\mathbf{U}$ by performing a single pass over the training set after the end of the training, as detailed in the Appendix of [Kristiadi et al., 2020] and clearly exemplified in the code-base of [Hobbhahn et al., 2021].

Let $\Sigma_k$ be the covariance matrix of the posterior over the last linear layer parameters for the k-th class obtained using the Laplace approximation around $\mu$, then, given an input $\mathbf{x}$, we obtain $\sigma_k = \phi(\mathbf{x})^\top \Sigma_k \phi(\mathbf{x})$ representing the variance of k-th logit $\mathbf{s}_k$. Once we obtain the covariance matrix, the Monte Carlo approximation of the predictive distribution (equation (4)) is obtained as:

$$\tilde{p} = \frac{1}{m} \sum_{i=1}^{m} \mathtt{softmax}(\mathbf{s}(i)), \tag{5}$$

where, $m$ logit vectors $\mathbf{s}(i)$ are sampled from a distribution with mean $\mathbf{s}$ and a covariance matrix (depending on the approximation used). Lu et. al [Lu et al., 2020] showed that similar performance can be achieved via the mean-field approximation which provides an approximate closed form solution of the integration in equation (4) involving the re-scaling of the logits and then taking the softmax of the re-scaled logit. The re-scaling is defined as follows:

$$\tilde{\mathbf{s}}_k = \frac{\mathbf{s}_k}{\sqrt{1 + \lambda \sigma_k^2}} \tag{6}$$

Note, the scaling of the k-th logit depends on its variance (obtained using the Laplace approximation) and a hyperparameter $\lambda$. This approximation is efficient in the sense that it does not require multiple samples as required in the MC approximation (which can become expensive as the number of classes and samples grow). In our experiments, we use the MC approximation, since we could not find an obvious way to fine-tune $\lambda$. Additionally, we observe that the mean-field approximation imposes a trade-off between calibration and OOD detection performance. Increasing $\lambda$, indeed, flattens the softmax distribution and improves OOD detection scores; although, as a consequence, harms calibration by making the network underconfident.

## F   Additional Insights: RegMixup encourages compact and separated clusters

Here we provide additional experiments to show that RegMixup encourages more compact and separated clusters in the feature space We use the well known Fisher criterion [Bishop, 2006, Chapter 4] to quantify the compactness and separatedness of the feature clusters.

**Fisher Criterion:** Let $\mathcal{C}_k$ denotes the indices of samples for $k$-th class. Then, the overall *within-class* covariance matrix is computed as $\mathbf{S}_W = \sum_{k=1}^K \mathbf{S}_k$, where $\mathbf{S}_k = \sum_{n \in \mathcal{C}_k} (\phi(\mathbf{x}_n) - \mu_k)(\phi(\mathbf{x}_n) - \mu_k)^\top$, $\mu_k = \sum_{n \in \mathcal{C}_k} \frac{\phi(\mathbf{x}_n)}{N_k}$, and $\phi(\mathbf{x}_n)$ denote the feature vector. Similarly, the *between-class* covariance matrix can be computed as $\mathbf{S}_B = \sum_{k=1}^K N_k (\mu_k - \mu)(\mu_k - \mu)^\top$, where $\mu = \frac{1}{N} \sum_{k=1}^K N_k \mu_k$, and $N_k$ is the number of samples in $k$-th class. Then, the Fisher criterion is defined as $\alpha = \texttt{trace}(\mathbf{S}_W^{-1} \mathbf{S}_B)$.

Note, $\alpha$ would be high when within-class covariance is small and between-class covariance is high, thus, *a high value of $\alpha$ is desirable*. In Figure 4, we compute $\alpha$ over the C10 dataset with varying degrees of domain-shift. As the amount of corruption increases, $\alpha$ gradually decreases for all the models. However, *RegMixup consistently provides the best $\alpha$* in most cases.

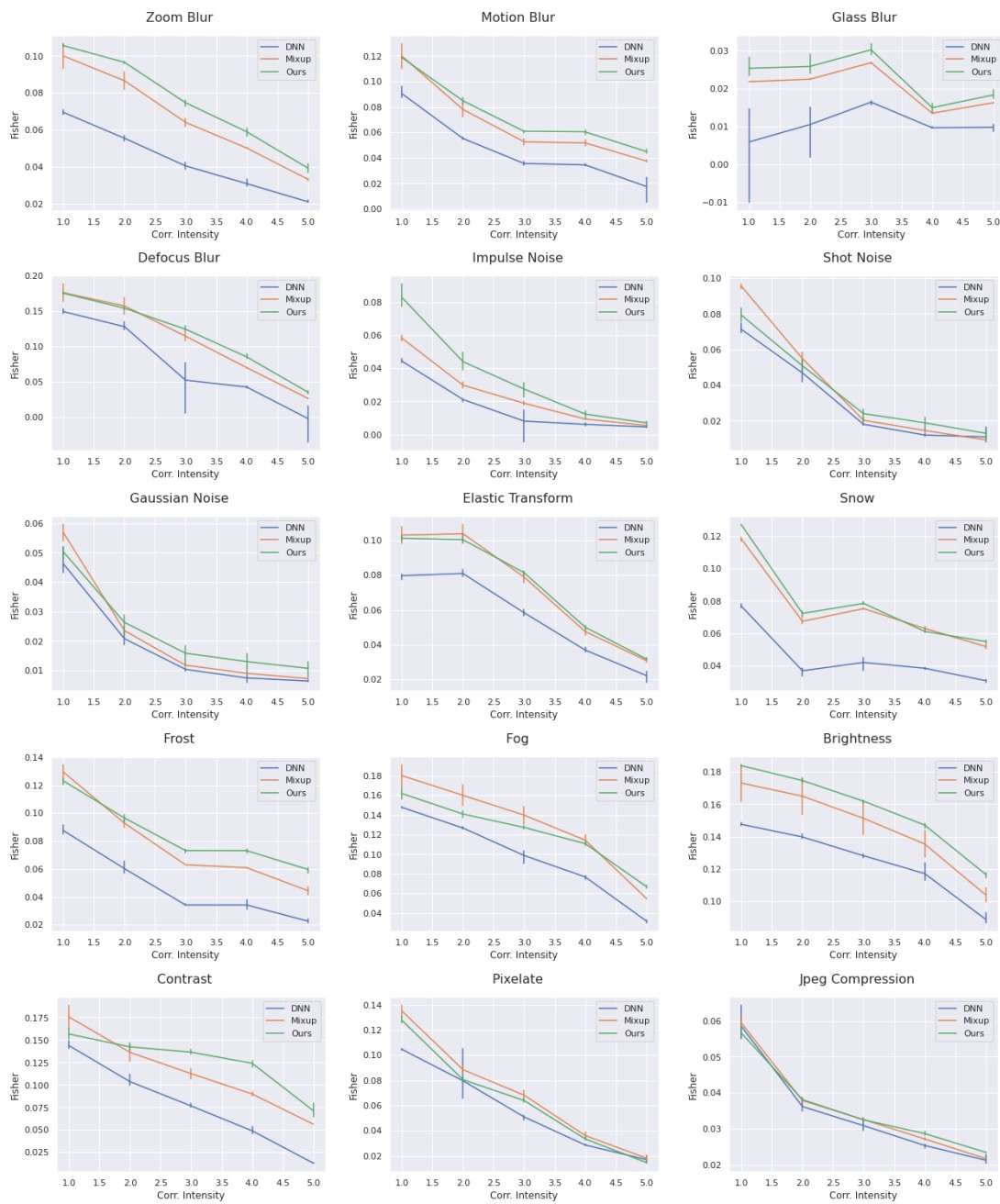

Figure 4: Fisher criterion for all the corruptions and intensity values of CIFAR-10-C (WRN28-10).

| | WRN28-10 | | RN50 | |
| | CIFAR10 (Val) | CIFAR100 (Val) | CIFAR10 (Val) | CIFAR100 (Val) |
|---|---|---|---|---|
| $\alpha$ | Accuracy | | | |
| 0.1 | 96.06 | 81.04 | 95.35 | 79.60 |
| 0.2 | 96.46 | 80.91 | 95.21 | 80.11 |
| 0.3 | 96.77 | 81.06 | 95.36 | 80.31 |
| 0.4 | 96.71 | 81.01 | 95.26 | 78.93 |
| 0.5 | 96.70 | 80.99 | 95.28 | 78.91 |
| 1 | 96.74 | 80.66 | 94.96 | 78.79 |
| 5 | 96.62 | 79.84 | 94.98 | 77.74 |
| 10 | 96.54 | 79.24 | 94.94 | 75.76 |
| 20 | 96.26 | 78.40 | 95.16 | 75.56 |

| | RN50 ImageNet |
|---|---|
| $\alpha$ | Accuracy |
| 0.1 | 77.10 |
| 0.2 | 77.02 |
| 1 | 76.19 |
| 10 | 72.17 |
| 20 | 71.51 |

Table 12: Mixup hyperparameter cross-validation: Accuracy (%) of WideResNet28-10 and ResNet50 on validation split of C10 and C100 for varying $\alpha$ values (table on the left), and of ResNet50 on ImageNet (table on the right).

| | DenseNet-121 | | PyramidNet-200 | |
| | CIFAR10 (Val) | CIFAR100 (Val) | CIFAR10 (Val) | CIFAR100 (Val) |
|---|---|---|---|---|
| $\alpha$ | Accuracy | | | |
| 0.1 | 95.89 | 80.54 | 96.71 | 82.34 |
| 0.2 | 96.10 | 80.80 | 96.70 | 82.17 |
| 0.3 | 96.21 | 80.80 | 96.67 | 81.70 |
| 0.4 | 96.06 | 79.71 | 96.79 | 82.62 |
| 0.5 | 95.98 | 80.17 | 96.92 | 81.90 |
| 1 | 96.07 | 79.08 | 96.89 | 81.80 |
| 10 | 95.93 | 75.76 | 96.69 | 79.50 |
| 20 | 95.74 | 76.03 | 96.60 | 78.75 |

Table 13: Mixup hyperparameter cross-validation: Accuracy (%) of DenseNet-121 and PyramidNet-200.

# G   ECE results

In this section we report the Expected Calibration Error (ECE) values. While ECE is more popular than AdaECE, the latter uses an adaptive binning scheme that better accounts for the bias introduced by the fact neural networks tend to concentrate most samples in high-confidence bins [Mukhoti et al., 2020].

| | WRN28-10 | | | | | | RN50 | | | | | |
| | CIFAR10 (Val) | | | CIFAR100 (Val) | | | CIFAR10 (Val) | | | CIFAR100 (Val) | | |
| | Accuracy | | | | | | | | | | | |
| $\alpha/\eta$ | 0.1 | 1 | 2 | 0.1 | 1 | 2 | 0.1 | 1 | 2 | 0.1 | 1 | 2 |
|---|---|---|---|---|---|---|---|---|---|---|---|---|
| 0.1 | 96.12 | 96.02 | 96.38 | 80.08 | 80.86 | 80.96 | 94.32 | 95.02 | 94.48 | 78.56 | 79.84 | 77.92 |
| 0.2 | 96.14 | 96.00 | 96.82 | 80.60 | 81.96 | 81.22 | 94.84 | 95.44 | 95.26 | 78.48 | 79.18 | 78.12 |
| 0.3 | 95.82 | 96.68 | 96.34 | 80.92 | 81.82 | 80.94 | 94.62 | 95.46 | 95.48 | 78.38 | 79.74 | 78.21 |
| 0.4 | 96.28 | 96.58 | 96.48 | 80.82 | 81.62 | 81.00 | 94.92 | 95.74 | 95.46 | 78.32 | 79.85 | 78.17 |
| 0.5 | 96.08 | 96.88 | 96.44 | 81.00 | 81.20 | 81.38 | 94.72 | 96.04 | 95.24 | 78,56 | 79.34 | 78.29 |
| 1 | 96.36 | 97.00 | 96.96 | 81.58 | 81.72 | 80.68 | 95.36 | 95.98 | 96.00 | 79.14 | 79.88 | 78.62 |
| 5 | 96.50 | 97.14 | 97.22 | 82.00 | 81.94 | 81.04 | 95.86 | 96.26 | 96.04 | 79.42 | 80.13 | 78.30 |
| 10 | 96.54 | 97.16 | 97.28 | 80.98 | 82.51 | 80.60 | 95.28 | 95.58 | 96.32 | 80.36 | 80.65 | 79.16 |
| 15 | 96.65 | 97.27 | 97.18 | 81.38 | 82.16 | 80.59 | 95.15 | 96.10 | 96.34 | 79.72 | 80.32 | 78.80 |
| 20 | 96.72 | 97.32 | 97.16 | 81.58 | 82.27 | 80.62 | 95.46 | 96.50 | 96.38 | 79.68 | 80.18 | 78.76 |
| 30 | 96.74 | 97.28 | 97.28 | 82.48 | 81.94 | 80.40 | 95.66 | 96.46 | 95.90 | 79.39 | 79.78 | 79.22 |

Table 14: RegMixup hyperparameter cross-validation: Accuracy (%) of WideResNet28-10 and ResNet50 on validation split of C10 and C100 for varying $\alpha$ and $\eta$ values.

| | **IND** | | | |
| | **WRN28-10** | | **RN50** | |
| | **C10 (Test)** | **C100 (Test)** | **C10 (Test)** | **C100 (Test)** |
| **Methods** | **ECE ($\downarrow$)** | | **ECE ($\downarrow$)** | |
|---|---|---|---|---|
| DNN | 1.26 | 3.88 | 1.38 | 3.05 |
| Mixup | 0.94 | 1.16 | **0.59** | 7.49 |
| RegMixup (**our**) | **0.62** | **1.65** | 0.62 | **1.51** |
| SNGP | 0.84 | 1.95 | - | - |
| Augmix | 1.67 | 5.54 | - | - |
| DE (5×) | 0.81 | 3.31 | 1.27 | 3.15 |

Table 15: CIFAR IND calibration performance (%).

| | **IND** | **Covariate Shift** | | | |
| | **ImageNet-1K (Test)** | **ImageNet-R** | **ImageNet-A** | **ImageNet-V2** | **ImageNet-Sk** |
| | **ECE ($\downarrow$)** | **ECE ($\downarrow$)** | **ECE ($\downarrow$)** | **ECE ($\downarrow$)** | **ECE ($\downarrow$)** |
|---|---|---|---|---|---|
| DNN | 1.85 | 13.52 | 44.91 | 4.17 | 14.51 |
| Mixup | **1.34** | 12.13 | 44.65 | 4.34 | 15.32 |
| RegMixup (**our**) | 1.38 | 13.32 | **41.23** | **3.41** | 15.37 |
| AugMix | 2.08 | **11.31** | 42.85 | 3.97 | **14.32** |
| DE (5×) | 1.39 | 13.59 | 42.92 | 4.07 | 17.34 |

Table 16: ImageNet calibration performance (%).

| | **Covariate Shift** | | | | | | | |
| | **WRN28-10** | | | | **R50** | | | |
| | **C10-C** | **C10.1** | **C10.2** | **C100-C** | **C10-C** | **C10.1** | **C10.2** | **C100-C** |
| **Methods** | **ECE ($\downarrow$)** | | | | **ECE ($\downarrow$)** | | | |
|---|---|---|---|---|---|---|---|---|
| DNN | 12.64 | 4.29 | 9.03 | 9.96 | 12.31 | 4.53 | 8.97 | 19.80 |
| Mixup | 7.54 | 4.28 | 7.40 | 10.32 | **10.23** | 5.59 | 10.59 | 13.57 |
| RegMixup (**our**) | 9.56 | **2.87** | **6.92** | **7.98** | 12.61 | **3.25** | **6.73** | **12.87** |
| SNGP | 11.33 | 4.32 | 8.68 | 10.45 | - | - | - | - |
| AugMix | **4.78** | 3.55 | 8.42 | 12.19 | - | - | - | - |
| DE (5 ×) | 10.11 | 2.98 | 7.53 | 12.43 | 13.12 | 4.32 | 7.10 | 12.53 |

Table 17: CIFAR CS calibration performance (%).

# H   Using other vicinal distributions as regularizers

In this Section we perform a preliminary analysis about using other techniques derived from Mixup as regularizers. In particular, we consider the following methods:

- CutMix Yun et al. [2019]: instead of performing the convex combination of two samples, cuts and pastes a patch of an image into another, and the area of the patch is proportional to the interpolation factor $\lambda$. We set the CutMix probability to $p = 1$ and cross-validate $\alpha \in \{0.1, 0.2, 0.3, 1, 10, 20\}$,
- RegCutMix: we replace Mixup with CutMix in our formulation. Same hyperparameters as above.
- Mixup + CutMix: we call this way the Transformers-inspired training procedure that at each iteration randomly decides (with equal probability) whether to apply Mixup or CutMix. For both Mixup and CutMix we cross-validate $\alpha \in \{0.1, 0.3, 1, 10\}$.
- RegMixup + CutMix: as above, but using Mixup as a regularizer of the cross-entropy when Mixup is selected. Hyperparameters as above, $\eta \in \{0.1, 1, 3\}$

| Methods | I.I.D. C10 | Covariate Shift C10-C | C10.2 | C10.1 | C100 | O.O.D. SVHN | T-ImageNet |
|---|---|---|---|---|---|---|---|
| | | Accuracy (↑) | | | | AUROC (↑) | |
| **WRN28-10** | | | | | | | |
| RegMixup | **97.46** | **83.13** | 88.05 | 92.79 | 89.63 | **96.72** | 90.19 |
| CutMix | 96.70 | 71.40 | 87.06 | 91.18 | 88.65 | 92.25 | **91.70** |
| RegCutMix | 96.79 | 71.26 | 86.84 | 92.10 | 89.24 | 89.47 | **91.68** |
| Vit-Mixup+CutMix | 97.23 | 77.60 | 87.85 | 92.30 | 89.23 | 83.46 | 90.86 |
| ViT-RegMixup+CutMix | 97.30 | 78.67 | 88.42 | 93.15 | 89.03 | 91.62 | 89.41 |
| ViT-RegMixup+RegCutMix | 97.47 | 79.10 | **88.79** | **93.35** | **89.93** | 94.18 | 90.39 |
| **R50** | | | | | | | |
| RegMixup | 96.71 | **81.18** | 86.72 | **91.58** | **89.63** | 95.39 | **90.04** |
| CutMix | 96.27 | 70.63 | 85.35 | 90.35 | 85.88 | 83.03 | 87.43 |
| RegCutMix | 95.85 | 70.22 | 84.64 | 90.31 | 85.10 | 84.76 | 87.36 |
| ViT-Mixup+CutMix | **96.87** | 76.36 | **87.05** | 90.75 | 89.19 | 93.39 | 89.76 |
| ViT-RegMixup+CutMix | 96.60 | 77.21 | 86.30 | 90.97 | 84.22 | 87.90 | 85.87 |
| ViT-RegMixup+RegCutMix | 96.47 | 75.36 | 86.01 | 91.28 | 86.36 | 87.21 | 88.42 |

Table 18: Accuracy and out-of-distribution detection performance (%) using other VRM techniques as regularisers for ResNet50 and WideResNet28-10 on CIFAR-10.

| Methods | I.I.D. C100 | Covariate Shift C100-C | C10 | O.O.D. SVHN | T-ImageNet |
|---|---|---|---|---|---|
| | | Accuracy (↑) | | AUROC (↑) | |
| **WRN28-10** | | | | | |
| RegMixup (Ours) | 83.25 | **59.44** | **81.27** | 89.32 | **83.13** |
| CutMix | 81.73 | 46.64 | 79.06 | 85.22 | 81.07 |
| RegCutMix | 82.30 | 47.37 | 80.99 | 84.42 | 81.85 |
| ViT-Mixup+CutMix | **84.05** | 54.94 | 80.81 | 85.36 | 81.32 |
| ViT-RegMixup+CutMix | 83.74 | 55.12 | 79.92 | 86.44 | 82.28 |
| ViT-RegMixup+RegCutMix | 83.92 | 55.50 | 80.75 | **90.23** | 83.07 |
| **R50** | | | | | |
| RegMixup | 81.52 | **57.64** | 79.44 | **88.66** | 82.56 |
| CutMix | 80.21 | 45.23 | 77.78 | 85.39 | 80.33 |
| RegCutMix | 79.07 | 44.64 | 77.56 | 80.52 | 79.18 |
| ViT-Mixup+CutMix | **82.39** | 56.40 | 80.53 | 81.09 | 81.01 |
| ViT-RegMixup+CutMix | 81.65 | 52.99 | 79.64 | 86.45 | 81.42 |
| ViT-RegMixup+RegCutMix | 82.10 | 53.57 | **80.66** | 84.43 | **83.03** |

Table 19: Accuracy and out-of-distribution detection performance (%) using other VRM techniques as regularisers for ResNet50 and WideResNet28-10 on CIFAR-100.

- RegMixup + RegCutMix: as above, but also using CutMix as a regularizer of the cross-entropy when CutMix is selected. Hyperparameters as above, $\eta$ is kept the same for both Mixup and CutMix.

The results are reported in Tables 18 and 19.

As it can be seen: (1) in most cases RegCutMix underperforms with respect to CutMix, (2) in most cases RegMixup outperforms CutMix and RegCutMix, (3) Mixup+CutMix represents an extremely competitive method for in-distribution accuracy, but not as competitive for distribution shift and out-of-distribution detection, (4) RegMixup+CutMix is most of the times inferior to Mixup+CutMix, (5) RegMixup+RegCutMix is an extremely competitive method in several cases. We leave to future research exploring whether and how it is possible to combine Mixup-inspired techniques

as regularizers to further improve the accuracy on i.i.d. and distribution-shifted data, and the out-of-distribution detection performance. We remark that in all these setups, RegMixup is extremely competitive and several times the best method, thus pointing out that despite its simplicity, it is extremely effective.