# OpenReview forum: "Using Mixup as a Regularizer Can Surprisingly Improve Accuracy & Out-of-Distribution Robustness"
_NeurIPS.cc/2022/Conference — NeurIPS 2022 Accept_

### Official Review · Reviewer_86F1 · 2022-07-03

**Rating:** 6
**Confidence:** 4
**Soundness:** 3 good
**Presentation:** 1 poor
**Contribution:** 3 good

**Summary:**

This paper proposes a simple change to Mixup data augmentation to also train with unperturbed images in addition to augmented images. Making this small change induces alpha cross-validation to choose stronger Mixup (closer to 50/50 image interpolation), yielding more image diversity without losing access to in-distribution samples during training. Extensive experiments show improvement in both in-distribution and out-of-distribution (OOD) accuracy, and OOD detection. This method avoids the common tradeoff of in-distribution and OOD accuracy.

**Questions:**

Do you have further or more concrete directions for future work in mind?

**Limitations:**

The authors do acknowledge a few experiments in which their method is outperformed, and partial explanations for why (e.g. AugMix).

**Strengths And Weaknesses:**

Strengths: The proposed method is very simple and easy to implement, as well as computationally inexpensive. It is validated with extensive experiments covering a wide set of models and datasets, and shows good results on multiple tasks.

Weaknesses: The paper does not read well due to many grammatical/writing issues. My first impression was negative because of this. As an example, the word "of" is missing in the title ("out of distribution"). Please copy edit the final version.

---

> ### Author Response · Authors · 2022-08-02
> **Thanks for appreciating the effectiveness & simplicity of RegMixup & the extent of our experiments. Below we seek to address the reviewers' concerns.**
>
> We thank the reviewer for acknowledging that RegMixup is simple, effective, inexpensive, and achieves more image diversity without losing access to in-distribution samples during training. We also thank the reviewer for appreciating the **extensive experiments** that show improvements in both in-distribution and out-of-distribution experiments without trading in-distribution accuracy for more robustness and reliability.
>
> _Below we address the raised concerns and hope for a constructive discussion phase_
>
> **Weakness: Grammatical/writing issues**
>
> **Ans** Thank you for your comment. The reason for not adding “of” in the title for Out of distribution was that it was _taking an extra line_. But as you suggested, we will make the necessary changes and update the draft. We would be grateful if the reviewer could point out the ones they identified so as to be sure to fix them all. We also improved the overall writing.
>
> **Limitation: The authors do acknowledge a few experiments in which their method is outperformed, and partial explanations for why (e.g. AugMix)**
>
> **Ans** Please note that not just RegMixup but also _all other approaches_ including Deep Ensembles is significantly outperformed by AugMix (please check Table 3 in the main paper) on particular set-up that **involves augmentations** similar to the ones used during the training of AugMix, hence such remarkable performance.
>
> However, in scenarios such as **natural distribution shift** where the shifts weren’t similar to the ones AugMix used during training, the RegMixup outperformed all the baselines.

---

> > ### Comment · Reviewer_86F1 · 2022-08-04
> > **Thanks for the response**
> >
> > Re: Writing
> >
> > Unfortunately I didn't keep track of all the writing issues, but I can give some more examples of a few writing changes I would make in the abstract and hopefully it gives a better sense of what I mean, that you can then apply to the rest of the paper. I would cut the word "being" in line 2, change "yield" to "yields" in line 7, clarify or reword "high-entropy" in line 9 to "high-uncertainty" (or somehow make it clear that by entropy you mean the entropy of the softmax predictions interpreted as discrete probabilities), and change "out-distribution" to "out-of-distribution" in line 10. These are small changes but they would make the paper flow better/easier to read.
> >
> > Re: Limitation
> >
> > I just want to clarify that the question in the reviewing form is whether or not the paper adequately discusses its own limitations; in my review I noted that you do already discuss some limitations (this is a good thing!).

---

> > > ### Author Response · Authors · 2022-08-04
> > > **Thank you for your suggestions to improve the writing and for encouraging the fact that we discuss the limitations of RegMixup**
> > >
> > > Great, we will definitely incorporate your comments to improve the writing quality. We will also make a careful pass to make sure that the paper flow is improved.
> > >
> > > Thank you for encouraging the fact that we mention the limitations of RegMixup in our paper. Appreciate it.
> > >
> > > Please let us know if you have any technical questions, we will be more than happy to get back to you.

---

### Official Review · Reviewer_FwuB · 2022-07-05

**Rating:** 6
**Confidence:** 4
**Soundness:** 3 good
**Presentation:** 4 excellent
**Contribution:** 3 good

**Summary:**

This paper presents RegMixup, which employs Mixup training as an additional regularize of the standard training. It can be treated as a special case of employing Mixup, but with a possibility of 0.5.  Some interesting findings are presented, for instance, RegMixup scales well with a beta distribution of (10, 10), and RegMixup provides high entropy barrier separating in-distribution from out-of-distribution samples. Extensive experiments are conducted on CIFAR-10, CIFAR-100, and ImageNet.  Calibration, OOD, and distribution/covariance shift results are presented. Overall, the presented method is simple yet insightful.

**Questions:**

My main concerns are listed in weaknesses, especially points 1 and 2, here is one more question/suggestion:

1: In Figure 2 and lines 108-109, authors claimed that Mixup employs a small value of α. But I believe many papers [1,2,3] employ beta (1,1) as the default setting. Authors may need to provide some supporting references or consider changing this claim

Overall I'm happy to change my rating to Borderline Accept if authors can strongly address weaknesses 1 and 2, where weakness 2 limits the application of RegMixup and weakness 1 limits the generality. If all of my concerns are well-addressed, I'm OK to raise my rating further into Weak Accept.

[1]: CutMix: Regularization Strategy to Train Strong Classifiers with Localizable Features,  ICCV 19.

[2]: You Only Cut Once: Boosting Data Augmentation with a Single Cut, ICML 22.

[3]: Lots of transformer papers and recent network architecture papers.

**Ethics Review Area:**

["I don’t know"]

**Limitations:**

Limitations are discussed. Negative societal impact are not discussed. A potential negative societal impact is that the interpretability of RegMixup is still limited and hence needs to be carefully applied to sensitive applications.

**Strengths And Weaknesses:**

Strengths:
1: The proposed method is simple, and can be easily used for training robust image classifiers.

2: Good presentation, well-written.

3: Experiments are extensive. Results are solid.


Weaknesses:
1: No experiments on CutMix (and other variants, but CutMix should be enough). Does RegMixup also help CutMix?

2: The training of RegMixup seems to see 2x samples per iteration. Thus the running speed is slow (as authors claimed 1.5 x slower). When compared to other methods, RegMixup seeing 2x samples may lead to unfair comparison.

3: Might be good to validate the performance of RegMixup on transformer and other training recipes (used in transformer models).

---

> ### Author Response · Authors · 2022-08-02
> **Thanks for appreciating the effectiveness & simplicity of RegMixup & finding our results solid. Below we seek to fully address the concerns (additional results are also presented)**
>
> We thank the reviewer for appreciating the **simplicity** of RegMixup and the **extensiveness and solidity of our results**. We also thank the reviewer for finding the paper **well written**.
>
> **Weakness 1: No experiments on CutMix, does RegMixup help CutMix?**
>
> **Ans**: Following your question and the comment from _Reviewer 2EUq_, we performed several new experiments to compare RegMixup with CutMix and a Transformer-inspired approach that uses a combination of Mixup and CutMix. We evaluate them in terms of in-distribution accuracy and CS/OOD robustness for proper comparison. Please **have a look at the reply [Comparisons with CutMix and ViT...| New Experiments](https://openreview.net/forum?id=5j6fWcPccO&noteId=uRqVr-3DiOm)**.
>
> We will add these additional experiments in our final draft. Thank you for raising this point, these experiments indeed have surely strengthened our work and showed its generality.
>
> **Weakness 2: RegMixup sees 2x samples per iteration, it might be unfair. This limits the applicability of RegMixup**
>
> **Ans**: We agree that RegMixup uses 2x samples per iteration, however, this does **not limit its applicability** at inference as the inference compute of RegMixup is exactly the same as a standard DNN with vanilla cross-entropy loss.
>
> Having said that, we would also like to bring **two widely used approaches** into the attention of the reviewer and discuss their compute requirements (we compare RegMixup with these approaches as well):
> - **Deep Ensembles (DE)**: One of the **best performing and most extensively used baselines** uses _5x data and 5x parameters_ during both training and inference. Still, the approach is widely accepted.
> - **AugMix**:  Similarly to our method, it processes a clean batch with the Cross-Entropy loss, and then processes _3 other augmented batches_ (not 1, like RegMixup), hence observing _4x the data_ (and taking _almost 4x training time_). Despite this, it is frequently used in the literature as a baseline.
>
> Please note, though we use 2x iterations per batch, we do not use additional datasets. Similarly to DE and AugMix, this should not be considered as a _major_ limitation of our approach.
>
> **Weakness 3: Might be good to validate the performance of RegMixup on transformer and other training recipes (used in transformer models).**
>
> **Ans**: Training transformers involves the usage of **multiple augmentations and tricks** at the same time, which interact in an unclear way, especially considering the difficulties of making transformers converge removing many of these techniques.
> - Techniques that work if used in isolation, damage the performance when combined with others in certain training regimes for the same architectures [7].
> - The fact they do not combine well with the current transformer training procedures **does not invalidate them**, especially considering CNNs are still extremely competitive architectures [2,5,6].
> - Identifying the optimal combination for transformer training is beyond the scope of our paper, and an interesting avenue for _future research_.
>
> However, following the suggestion of comparing with transformer-inspired training techniques (also commented by _Reviewer 2EUq_), we did perform **additional experiments** and compare RegMixup with CutMix and transformer-inspired Mixup-CutMix combination. Please check the _New Experiment 2_ in our **comment [Comparisons with CutMix and ViT...| New Experiments](https://openreview.net/forum?id=5j6fWcPccO&noteId=uRqVr-3DiOm)**
>
> **Q1: In Figure 2 and lines 108-109, authors claimed that Mixup employs a small value of α. But I believe many papers [1,2,3] employ beta (1,1) as the default setting. Authors may need to provide some supporting references or consider changing this claim.**
>
> **Ans**: We provide a **thorough analysis** to support our argument. Please **check the results [Analyzing the value alpha in Mixup ...](https://openreview.net/forum?id=5j6fWcPccO&noteId=T2epISSdoMm)**.  You may also find the **reply to Q3 to Reviewer 2EUq** more insightful for this question.
>
> [1] mixup: Beyond Empirical Risk Minimization, Zhang et al. ICLR 2018
>
> [2] Resnet strikes back: an improved training procedure in timm, Wightman et al. ImageNet PPF Workshop NeurIPS 2021
>
> [3]: CutMix: Regularization Strategy to Train Strong Classifiers with Localizable Features, ICCV 19.
>
> [4]: You Only Cut Once: Boosting Data Augmentation with a Single Cut, ICML 22.
>
> [5] A Convnet for the 2020s, Liu et al. CVPR 2022
>
> [6] An Impartial Take to the CNN vs Transformer Robustness Contest, Pinto et al.  Arxiv 2022
>
> [7] How to train your ViT? Data, Augmentation, and Regularization in Vision Transformers, Steiner et Al. 2022 TMLR

---

> > ### Comment · Reviewer_FwuB · 2022-08-03
> > **Response to authors 9722**
> >
> > Thank you for preparing the rebuttal.
> >
> > Weakness 1:Oh I'm not asking for a comparison between RegMixup and CutMix. I'm interested in seeing a method like "RegCutMix". It would be great to see whether CutMix also behaves as a good regularizer.
> >
> > Weakness 2: RegMixup uses x2 samples, and costs x 1.5 training time. I would suggest adding an experiment to address this. Let's say, we can train RegMixup for 100 epochs, but train other baselines (Mixup) for 200 epochs and 150 epochs.
> >
> > Weakness 3: Again I'm not suggesting adding comparisons. My point is, if replacing Mixup with RegMixup in such transformer training recipes, what will happen?
> >
> > Q1: Great. This addresses my concern.
> >
> > Overall my concerns are not strongly addressed yet. I would suggest authors perform some experiments on weaknesses 1-3 mentioned above.

---

> > > ### Author Response · Authors · 2022-08-03
> > > **Experimental results addressing Weakness 2 | Observations on Weaknesses 1 and 3 and mention ongoing experiments**
> > >
> > > **Weaknesses 1 and 3, answer**: Thanks for the clarification. Yes, it would definitely be interesting to see how an approach similar to RegMixup improves CutMix (or say label smoothing).
> > >
> > > - We have mentioned this as a future exploration in the paper.
> > > - We hope that the reviewer would agree that trying different variants of Mixup (4K+ citations) would be very challenging. However, following your suggestion, we are **currently running** these experiments and will upload the results as soon as they are available.
> > >
> > > We would like to emphasize that not providing experiments for the suggested variant of CutMix (RegCutMix) should not undermine the generality and usefulness of our approach. As we have shown, **RegMixup is already outperforming CutMix or Transformer-inspired techniques in many experiments**.
> > >
> > > **Ongoing Experiments**:
> > > - RegCutMix – CutMix as a regularizer.
> > > - Transformer-inspired experiments where Mixup is replaced by RegMixup.
> > >
> > > **Weakness 2, answer**: Thank you for your comment. We have performed experiments in this regard. Considering RegMixup is trained for 350 epochs, we train **DNN** and **Mixup** for **700 epochs** (adapting the scheduler accordingly). The results are the following (e350 and e700 denote models trained for 350 and 700 epochs, respectively).
> > >
> > > For CIFAR-10:
> > > |                                | IND             | DS              |               |               | OOD   |       |               |
> > > |--------------------------------|-----------------|-------------------|---------------|---------------|-------|-------|---------------|
> > > |                                | CIFAR10     | CIFAR-C       | CIFAR10.2         | CIFAR10.1         | CIFAR100 | SVHN  | Tiny-ImageNet |
> > > |                                | Acc     | Acc       | Acc   | Acc   | AUROC | AUROC | AUROC         |
> > > | R50                           |                 |                   |               |               |             |               |
> > > | RegMixup (Ours) e350  | **96.71**   | **81.18**    | **86.72** | **91.58** | **89.63** | **95.39** | **90.04**         |
> > > | DNN e350   | 95.19  | 75.18 | 83.31 | 88.58 | 88.61 | 93.20 | 87.82 |
> > > | DNN e700   | 95.51   | 75.99    | 83.55 | 88.90 |  87.23 | 88.53 | 86.03         |
> > > | Mixup e350 | 96.05 | 78.63 | 84.61 | 90.03 | 84.24| 89.40 | 84.89 |
> > > | Mixup e700 | 96.40   | 78.16    | 84.80 |  90.45 | 77.90 | 80.51 | 78.85         |
> > > | WRN                          |                 |                   |               |               |             |               |
> > > | RegMixup (Ours) e350 | **97.46**   | **83.13**  | **88.05** |  **92.79** | **89.63** | **96.72** | **90.19**         |
> > > |DNN e350 | 96.14 | 76.60 | 84.79 | 90.73| 88.61 | 96.00 | 86.44|
> > > | DNN e700   | 96.28   | 77.21    | 84.60  | 90.60 | 88.52 | 94.64 | 87.39         |
> > > | Mixup e350| 97.01 | 81.68 | 86.55 | 91.29 | 83.17 | 87.53 | 84.02|
> > > | Mixup e700 | 97.26   | 81.79    | 87.10  | 91.80 | 76.66 | 84.31 | 77.13         |
> > >
> > >
> > > For CIFAR-100:
> > > |                 |    IND   |      DS | OOD   |       |               |
> > > |-----------------|:--------:|--------:|-------|-------|---------------|
> > > |                 | CIFAR100 | CIFAR-C | CIFAR-10 | SVHN  | Tiny-ImageNet |
> > > | R50             | Acc      | Acc     | AUROC | AUROC | AUROC         |
> > > | RegMixup (Ours) e350 |   **81.52**  |   **57.64** | **79.44** | **88.66** | **82.56**         |
> > > | DNN e350 | 79.19 | 50.62 | 79.33 | 82.45 | 79.89 |
> > > | DNN e700        |   80.37  |   51.67 | 78.93 | 81.83 | 80.10         |
> > > | Mixup e350 | 80.12 | 53.96 | 77.02 | 76.86 | 80.14 |
> > > | Mixup e700      | 81.26    | 56.43   | 77.45 | 85.52 | 80.15         |
> > > | WRN             |          |         |       |       |               |
> > > | RegMixup (Ours) e350 | **83.25**    |**59.44**   |**81.27** | **89.32** | **83.13**         |
> > > | DNN e350 | 81.58 | 52.54| 81.06 | 79.68 | 80.99 |
> > > | DNN e700        | 81.94    | 52.75   | 80.66 | 83.86 | 82.29         |
> > > | Mixup e350 | 82.60 | 56.99 | 78.37 | 78.68 | 80.61 |
> > > | Mixup e700      | 82.23    | 57.40   | 78.85 | 84.90 | 78.71         |
> > >
> > > **Observations**:
> > >
> > > These experiments clearly show that:
> > > - **RegMixup outperforms the baselines** even when they are trained for **twice the number of epochs**
> > > - Training the baselines for more iterations does improve their in-domain accuracies, but can sometimes improve and sometimes degrade the accuracy under covariate shift, and most times degrades the out-of-distribution detection properties. Perhaps this is caused by overfitting, further investigation in this regard is not within the scope of the submitted work.

---

> > > > ### Comment · Reviewer_FwuB · 2022-08-03
> > > > **Response to authors 9722**
> > > >
> > > > Thanks for your quick reply.
> > > >
> > > > Weaknesses 1 and 3: I agree with the authors that there is no need to try many variants of Mixup. So my suggestion is to test "RegCutMix" only (As CutMix is the most notable variant). For weakness 3, it's also good to see whether RegMixup is suitable for complex training recipes (many tricks only work for simple recipes).  Looking forward to hearing new results! My main concern is weakness 1. It's fine if RegMixup does not perform very well with complex training recipes.
> > > >
> > > > Weakness 2: 700 epochs might be too much for CIFAR datasets. How about 200 epochs for RegMixup and 350 epochs for Mixup (so you don't have to re-run experiments for Mixup)?

---

> > > > > ### Author Response · Authors · 2022-08-07
> > > > > **New experiments and findings for Weakness 3 | Hopefully these experiments will be satisfying**
> > > > >
> > > > > Finally we have some answers to your questions. Hopefully you will appreciate the efforts and the findings.
> > > > >
> > > > > Over the last couple of days, we *trained* nearly **300 models** to further answer all your questions. We ourselves were also excited to know the answers. Though we could not manage to try all possible hyperparameters because of the lack of time, we believe that we still have satisfying conclusions.
> > > > >
> > > > > Hyperparameters involved:
> > > > > - RegMixup = CE + $\eta$ Mixup. Therefore, $\eta$, and $\alpha$ for the Beta distribution
> > > > > - CutMix. $\alpha$ for the Beta distribution
> > > > >
> > > > > Models: WideResNet28-10 and ResNet50
> > > > >
> > > > > Datasets: CIFAR10 and CIFAR100
> > > > >
> > > > > **Weakness 3**: Let us start with this one (**Impact of RegMixup on ViT type complex training**)
> > > > >
> > > > > We tried the following set of experiments:
> > > > >
> > > > > **Experiment 1: ViT-RegMixup-CutMix** – Here we replaced the standard Mixup with our RegMixup.
> > > > >
> > > > > Algo: If boolean == 1, CE + $\eta$ Mixup, otherwise, CutMix
> > > > >
> > > > > Hyperparam: $\eta$ = [0.1,1,3], $\alpha_{regmixup}$ = [0.3, 1, 10], $\alpha_{cutmix}$ = [0.2, 0.4, 1, 20]
> > > > >
> > > > > **Experiment 2: ViT-RegMixup-RegCutMix** – Here we replaced the standard Mixup with our RegMixup and also CutMix with RegCutMix.
> > > > >
> > > > > Algo: If boolean == 1, CE + $\eta$ Mixup, otherwise, CE + $\eta$ CutMix
> > > > >
> > > > > Hyperparam: $\eta$ = [0.1,1,3], $\alpha_{regmixup}$ = [0.3, 1, 10], $\alpha_{cutmix}$ = [0.2, 0.4, 1, 20]
> > > > >
> > > > > **CIFAR10 results**
> > > > >
> > > > >
> > > > > | | IND| DS | | | OOD   |  | |
> > > > > |-----|---|---|---|---|--|-|----|
> > > > > | | CIFAR10| CIFAR-C| CIFAR10.2| CIFAR10.1| CIFAR100 | SVHN  | Tiny-ImageNet |
> > > > > || Acc| Acc| Acc   | Acc   | AUROC | AUROC | AUROC|
> > > > > | **R50** |||||||
> > > > > | RegMixup (Ours) | 96.71   | **81.18**     | 86.72 |  **91.58** | **89.63** | **95.39** | 90.04|
> > > > > | ViT-Mixup+CutMix  | 96.81  | 75.50  | 86.10 | 91.05  | 89.28 | 95.10 | 89.50|
> > > > > |ViT-RegMixup-CutMix|  96.68 | 74.60 | **87.20** | **91.60** | 88.31 | 95.19 | **90.38** |
> > > > > |ViT-RegMixup-RegCutMix| **97.01** | 77.88 | 86.90 | 91.55 | 88.64 | 93.00 | 89.52 |
> > > > > | **WRN**    | | || | ||
> > > > > | RegMixup (Ours) | 97.46   | **83.13**  | 88.05 |  92.79 | 89.63 | **96.72** | **90.19**|
> > > > > |ViT-Mixup+CutMix | 97.58   |  76.49   | 87.70   | 92.75  | 85.75 | 92.66 | 86.77|
> > > > >  |ViT-RegMixup-CutMix| 97.36 | 79.95 | **88.70** | **93.45** | 89.27 | 89.57 | 89.92|
> > > > > |ViT-RegMixup-RegCutMix| **97.63** |77.87|  88.50|  94.15 | **89.79** | 90.55 | 89.69|
> > > > >
> > > > > **CIFAR100 results**
> > > > >
> > > > > |  |IND|DS | OOD|||
> > > > > |---|:--:|--:|--|-------|--|
> > > > > | | CIFAR100 | CIFAR-C | CIFAR10 | SVHN  | Tiny-ImageNet |
> > > > > | **R50**| Acc| Acc| AUROC | AUROC | AUROC|
> > > > > | RegMixup (Ours)  |   81.52  |   **57.64** | 79.44 | **88.66** | **82.56**|
> > > > > | ViT-Mixup+CutMix |   81.98  |   52.84 | 80.57 | 86.77 | 80.79|
> > > > >  |ViT-RegMixup-CutMix | 81.92 | 53.47 | 79.97 | 86.81 | 81.63 |
> > > > > |ViT-RegMixup-RegCutMix| **82.50** | 53.72  | **81.25** | 82.39 | 82.53|
> > > > > | **WRN**|||| ||
> > > > > | RegMixup (Ours)  | 83.25    | **59.44**   | **81.27** | 89.32 | **83.13**|
> > > > > | ViT-Mixup+CutMix | 83.87   | 54.07   | 79.20 | 89.99 | 81.01|
> > > > >  |ViT-RegMixup-CutMix | 83.90 | 55.23 |  79.69  | 84.34  | 81.85 |
> > > > > |ViT-RegMixup-RegCutMix| **84.11** | 56.11 | 80.42 | **90.77** | 82.57|
> > > > >
> > > > > Conclusions:
> > > > > - Using the **regularized version of Mixup (or CutMix) for ViT inspired training does help** in improving the accuracy (marginally)
> > > > > - The DS and OOD performance is also improved in many cases but is degraded in a few as well.
> > > > > - Though using a regularized version is quite promising, **RegMixup alone is still the best performing one (overall) and much more stable across all the benchmarks**.
> > > > > - More hyperparameters cross-validation might improve things further.

---

> > > > > > ### Author Response · Authors · 2022-08-07
> > > > > > **New Experiments for Weakness 1 and 2 | Hopefully our efforts and new findings will further validate the effectiveness of RegMixup**
> > > > > >
> > > > > > Here we provide answers to **Weaknesses 1 and 2**. As mentioned, we trained hundreds of models over the last couple of days to reach satisfying conclusions. The hyperparameters chosen are based on our experience regarding what was working the best, however, a more extensive search might be needed for solid conclusions.
> > > > > >
> > > > > > **Weakness 1: RegCutMix**
> > > > > >
> > > > > > Obj: CE + $\eta$ CutMix
> > > > > >
> > > > > > Hyperparam: $\eta$ = {0.1,1,3} and $\alpha$ = {1,10,20,30}
> > > > > >
> > > > > > **CIFAR10 Results**:
> > > > > >
> > > > > > |  | IND  | DS  | | | OOD   | | |
> > > > > > |-|-|-|--|--|--|--|--|
> > > > > > | | CIFAR10| CIFAR-C| CIFAR10.2| CIFAR10.1| CIFAR100 | SVHN  | Tiny-ImageNet |
> > > > > > |  | Acc     | Acc       | Acc   | Acc   | AUROC | AUROC | AUROC|
> > > > > > | **R50** |||||||
> > > > > > | RegMixup (Ours)| **96.71**   | **81.18**     | **86.72** |  **91.58** | **89.63** | **95.39** | 90.04         |
> > > > > > | CutMix   | 96.51   | 71.09     | 85.25 | 90.50 |  86.90 | 86.62 | **90.30**         |
> > > > > > |RegCutMix| 96.32 | 69.30 | 85.00 | 90.10 | 78.44 | 75.10 | 80.38 |
> > > > > > | **WRN**|||||||
> > > > > > | RegMixup (Ours) | **97.46**   | **83.13**  | **88.05** |  92.79 | **89.63** | **96.72** | **90.19**         |
> > > > > > | CutMix| 97.15| 73.39| 86.85  | 92.55 | 88.64 | 91.11 | 88.05|
> > > > > > |RegCutMix| 97.08 | 72.82 | 86.80 | **92.85** | 81.35 | 86.38 | 81.78|
> > > > > >
> > > > > > **CIFAR 100 results**:
> > > > > >
> > > > > > ||IND|DS | OOD   |||
> > > > > > |-|:---:|---:|-|-|--|
> > > > > > | | CIFAR100 | CIFAR-C | CIFAR10 | SVHN  | Tiny-ImageNet |
> > > > > > | **R50**| Acc| Acc| AUROC | AUROC | AUROC|
> > > > > > | RegMixup (Ours) |   **81.52**  |**57.64** | **79.44** | **88.66** | **82.56**|
> > > > > > | CutMix|   81.30  |   48.66 | 78.92 | 81.40 | 80.15|
> > > > > > |RegCutMix | 79.36 | 44.70  | 78.44 | 75.10 | 80.38|
> > > > > > | **WRN**||||||
> > > > > > | RegMixup (Ours) | **83.25**| **59.44**   | 81.27 | **89.32** | **83.13**|
> > > > > > | CutMix| 83.20    | 51.16| 82.19 | 88.78 | 82.32|
> > > > > > |RegCutMix| 82.56 | 46.85 | **82.28** | 88.73 | 82.42 |
> > > > > >
> > > > > > Conclusions:
> > > > > > - In our experiments, we did **not** find **RegCutMix to outperform CutMix** (except in 3 cases)
> > > > > > - **RegMixup** (our) overall is again **better than CutMix and RegCutMix** both.
> > > > > > - Perhaps more hyperparameter search such as lower values of $\alpha$, or different values of $p$ (deciding whether to apply CutMix or not) for CutMix would reveal more promising numbers.
> > > > > > - Please note, though RegCutMix did not provide promising results (yet), using regularized version of Mixup and CutMix **did provide promising results** for ViT-inspired training as we showed in [Weakness 3 experiments](https://openreview.net/forum?id=5j6fWcPccO&noteId=jJzb_mmOCPm)
> > > > > >
> > > > > > **Weakness 2: Lower number of epochs for RegMixup**
> > > > > > - We did evaluate RegMixup trained with a lower number of epochs, as suggested. Its performance degraded compared to the performance when training for 350 epochs.
> > > > > > - This observation is not that surprising as lowering down the number of epochs has a huge impact on the optimal choice of many other hyperparameters (learning rate, scheduler milestones etc etc.) therefore a **proper cross-validation is needed to conclude something** here.
> > > > > > - Standard DNN and Mixup did not strongly benefit from longer training epochs either.
> > > > > > - Therefore, even if RegMixup is slower to train compared to standard DNN, this is not a significant bottleneck as it is **still much more efficient than many other widely used models such as Deep Ensemble and AugMix**.
> > > > > > - Also, **at test time it is as efficient as standard DNN**. Therefore, we would request the reviewer to not see it as a big disadvantage. The **gains** obtained with such a simple approach on a wide variety of tasks clearly **outweigh this particular disadvantage**.
> > > > > >
> > > > > > **Remarks**: We hope that we have done enough experiments to answer all the queries of the reviewer in such a short span of time. The questions raised indeed were very interesting and the new experiments performed clearly show, again, that RegMixup is extremely **simple** and **highly effective**.
> > > > > >
> > > > > > Please note that the methods proposed in the literature (discussed in the paper) not just significantly degrade the accuracy while improving the robustness, they also require modifications to the architecture etc. RegMixup does not require any of that: it is super simple, intuitive, and highly effective.
> > > > > >
> > > > > > Thank you once again for the constructive interactions we are having. Looking forward to reading your comments.

---

> > > > > > > ### Comment · Reviewer_FwuB · 2022-08-08
> > > > > > > **Response to Authors 9792**
> > > > > > >
> > > > > > > Thank you for addressing my concerns.
> > > > > > >
> > > > > > > Weakness 1: Though RegCutMix does not show promising results yet, it's great to see RegMixup + RegCutMix works. This does address my main concern. In the camera-ready version, I would suggest authors add an experiment on ImageNet dataset (both CNN and ViT), to see whether RegCutMix+RegMixup works in larger datasets.
> > > > > > >
> > > > > > > Weakness 2: I understand training for 200 epochs can degrade the results. Though I feel efficiency is important for data augmentation/ training tricks, it's ok to present such simple yet interesting findings in the Neurips community.
> > > > > > >
> > > > > > > Weakness 3: Great! Authors did more than my expectations.
> > > > > > >
> > > > > > > Most of my concerns are addressed. I will raise my rating to weak accept.

---

> ### Author Response · Authors · 2022-08-08
> **Many thanks for appreciating our efforts, having great interaction, suggesting valuable experiments, and raising your score**
>
> Dear Reviewer,
>
> We would like to thank you for your valuable time and highly relevant experiments.
>
> We aren't sure whether our paper will be accepted or not, but the way you made this reviewer-author discussion-phase constructive and interesting is remarkable. Thank you for making it such a positive experience.
>
> Your suggested experiments and new results have certainly improved the quality of our work and further validated the effectiveness and usefulness of RegMixup. Thank you for that. We will include these new results in our revised draft. We will also try to include the ImageNet experiments as per your suggestion.
>
> Thank you once again for your time and suggestions!

---

> > ### Comment · Reviewer_FwuB · 2022-08-08
> > **Response to Authors 9792**
> >
> > Dear Authors,
> >
> > Thanks for your kind words. It's glad to see my suggestions are somehow valuable for improving your manuscript. I also appreciate your efforts in this discussion period. Your rebuttal/response is very well prepared.
> >
> > Best luck with your submission!

---

### Official Review · Reviewer_2EUq · 2022-07-12

**Rating:** 6
**Confidence:** 3
**Soundness:** 2 fair
**Presentation:** 3 good
**Contribution:** 2 fair

**Summary:**

This paper proposes a simple modification of the well-known Mixup that uses the combination of vanilla training loss and Mixup training loss as the training objective. This modification helps to overcome the two issues of Mixup: 1) Mixup chooses extremely small $\alpha$ during cross-validation and 2) Mixup can not distinguish in-distribution and out-of-distribution samples. Experiment results show that RegMix outperforms Mixup and many other baselines in in-distribution accuracy, Covariate Shift accuracy, and out-of-distribution detections.

**Questions:**

1. This paper states that one of the limitations of Mixup is that it uses small $\alpha$ for good generalization. And this paper uses $\alpha=0.3$ for CiFAR-10 & CIFAR-100, and $\alpha=0.1$ for ImageNet. But as I know in the case of image classification, e.g., CIFAR-10, ImageNet-1k, the commonly used $\alpha$ is 1 which leads to a uniform distribution. Could the authors provide results of these datasets under the commonly used $\alpha$ and provide some explanation on why Mixup prefers small $\alpha$ in this case? I believe this will provide a more solid sound.

2. Since the choice of $\alpha$ is the limitation of MIxup and benefit of RegMix, I would suggest putting the $\alpha$ used for these two methods in the main body instead of in the Appendix.

3. (Minor one) It would be better to provide the comparison of a failure case of Mixup and a success case of RegMix for OOD.

**Limitations:**

1. Mixup was developed in 2018 and several better mixup-based methods have been proposed, e.g. the common training pipeline of ViT now uses a combination of Mixup and CutMix. I was wondering how these methods perform against CS and OOD and if RegMix together with these new mixup-based methods can still lead to better results.

2. It would be better to provide some theoretical justifications to show why RegMix can help overcome the issues of Mixup. Specifically, why Mixup prefers small $\alpha$ but RegMix prefers a large one.

**Strengths And Weaknesses:**

Strengths:
1. This paper suggests a simple modification of Mixup that fixes the two issues of Mixup and leads to better CS and OOD performance.
2. The paper is well-written. It first defines the issues of Mixup and shows how RegMix is able to overcome these issues.
3. Extensive experiments have been carried out on various image datasets and several tasks.

Weaknesses:
1. The proposed method is not that different from the original Mixup.
2. The proposed method is designed for the earliest mixup-based method. And we are not sure if it also works for the recent progress on mixup-based methods, e.g., CutMix, PuzzleMix.
3. There is no theoretical justification for why RegMix works.

---

> ### Author Response · Authors · 2022-08-02
> **Thanks for appreciating the effectiveness and simplicity of RegMixup. Below we seek to fully address the concerns (additional results are also presented)**
>
>
> We thank the reviewer for appreciating that RegMixup is a **simple fix** to the difficulty of Mixup in distinguishing in-distribution and out-of-distribution samples, and **outperforms** it on an **extensive set of datasets and tasks**. We also thank the reviewer for finding the paper **well written**.
>
> Below we answer the raised concerns and hope for a constructive discussion phase.
>
> **Weakness 1: The proposed method is not that different from the original Mixup.**
>
> **Ans:** We agree with the reviewer, our modification is extremely simple and might even sound very obvious. However, on a variety of experiments we show RegMixup to be **highly effective**, specifically much better than Mixup on OOD detection problems. To summarize:
> - We believe that the **simplicity** of RegMixup should be considered as its **strength**.
> - The final simplified form of RegMixup is very well backed by the **theoretical justification** (the underlying approximations to the Vicinal Risk Minimization) and extensive experiments show its **effectiveness**.
>
> These conceptual and behavioral differences differentiate RegMixup from Mixup.
>
> **Weakness 2 (and Limitation 1): The proposed method is designed for the earliest mixup-based method. And we are not sure if it also works for the recent progress on mixup-based methods e.g., CutMix PuzzleMix; …e.g. the common training pipeline of ViT now uses combination of mixup and CutMix. …”**
>
> **Ans**: Following your question, we performed several new experiments to **compare RegMixup with CutMix and a Transformer-inspired approach** that uses a combination of Mixup and CutMix. We evaluate them in terms of in-distribution accuracy and CS/OOD robustness for proper comparison. Please **have a look at the reply [Comparisons with CutMix and ViT...| New Experiments](https://openreview.net/forum?id=5j6fWcPccO&noteId=uRqVr-3DiOm)**.
>
> We will add these additional experiments in our final draft. Thank you for bringing this point forward, these experiments indeed have strengthened our work and showed its generality.
>
>
> **Weakness 3 (Limitation 2): There is no theoretical justification for why RegMix works;
> “..theoretical justifications to show…Specifically, why Mixup prefers small \alpha but RegMixup prefers a large one.”**
>
>
> **Ans**: We believe that we have provided a proper theoretical justification behind RegMixup **in Section 2** of the paper. Similarly to Mixup, our justification is **based on Vicinal Risk Minimization (VRM)**, however, we also properly discuss why the approximations used in VRM might lose VRM’s capabilities and why it is important to have a better approximation to the vicinal distribution. Based on this justification, we built RegMixup as a simple combination of ERM and VRM.
>
> We also show that how large $\alpha$ in RegMixup is equivalent to a **proxy to maximizing entropy** and hence we justify why RegMixup is relatively **more uncertain outside the data distribution**. Thus, improved performance on OOD tasks.
>
> We also mention that the preference of large \alpha in RegMixup is possible as due to the **cross-entropy loss over clean samples**, the danger of data-shift due to large \alpha in regular Mixup does not exist anymore. The model has more freedom to explore a large range of \alpha.
>
> We would really appreciate it if the reviewer could mention what more insights and justifications are expected from us. We will be more than happy to discuss them.
>
> **Q1 This paper states … Could the authors provide results of these datasets under the commonly used \alpha and provide some explanation on why Mixup prefers small \alpha in this case? I believe this will provide a more solid sound.**
>
> **Ans**: Thank you for this question. We provide **thorough cross-validation** of $\alpha$ and justifications behind why few approaches use $\alpha$ = 1 in our **reply [Analyzing the value alpha in Mixup ...](https://openreview.net/forum?id=5j6fWcPccO&noteId=T2epISSdoMm)**
>
> We hope these results will be convincing enough.
>
>
> **Q3: (Minor one) It would be better to provide the comparison of a failure case of Mixup and a success case of RegMix for OOD.**
>
> Could you please **clarify** this point? Are you asking to show some images that RegMixup correctly classifies as ind/ood and Mixup does not as anecdotal evidence of the better performance of RegMixup? (We did not include this type of evidence as it is anecdotal, while the AUROC produces a quantitative description on the whole test sets, but we can update the draft accordingly if anecdotal evidence might help convincing to the readers).

---

> > ### Comment · Reviewer_2EUq · 2022-08-05
> > **Thanks for the detailed response**
> >
> > Thanks for the detailed response. And thanks for the extensive experiments on $\alpha$. While the novelty concern remains, most of my concerns are addressed. I appreciate your clarifications. I will increase my rating to 6.
> >
> > Q3: (Minor one) Yes, I suggested showing some images like Fig.2 and 3 and their logits from models trained by RegMixup and vanilla pipeline.

---

> > > ### Author Response · Authors · 2022-08-05
> > > **Many thanks for appreciating our efforts, new analysis, and raising your score**
> > >
> > > Dear Reviewer,
> > >
> > > We thank you for reading our replies and appreciating the effort.
> > >
> > > Regarding Q3: (Minor one), we will update our draft with more images like Fig 2 and 3 (as you suggested) for better clarity.
> > >
> > > We will also include our additional analysis on $\alpha$ and experiments in our revised draft.
> > >
> > > Thank you once again for your time and valuable suggestions.

---

> ### Author Response · Authors · 2022-08-04
> **Gentle nudge | We request you to let us know if there is anything that still requires clarification**
>
> Dear Reviewer,
>
> We hope that we have satisfactorily replied to all your concerns. Our reply is supported heavily by new experiments and conceptual/theoretical justifications.
>
> We understand that you must be super occupied so apologies for the reminder. We request you to have a look at our reply and let us know if there is anything that isn't clear yet. We will be happy to get back to you to clarify any doubts or questions you may still have.
>
> Looking forward to a constructive discussion.
>
> Thanks!

---

### Author Response · Authors · 2022-08-02
**Comparisons with CutMix and ViT inspired Mixup-CutMix combination | New Experiments**

Before we begin discussing additional experiments to show that RegMixup is still the best performing one in most cases and provides robust models, we would like to emphasize that
- We have already performed **extensive experiments** (hundreds of them) and compared our approach with several baselines
- Recent approaches such as SNGP, DUQ etc. _do not consider Mixup_ as one of the baselines
- We not only add Mixup and AugMix as additional baselines, we extensively compare RegMixup with many recent approaches such as DUQ, Deep Ensemble, DNN-SRN, DNN-SN, KFAC-LLLA (Bayesian at test time) on a variety of datasets (CIFAR10/100, ImageNet) using both ResNet and WideResNet.

Since our theme is robustness, we believe that we have covered many recent relevant approaches along with a few additional ones as baselines and have provided significant empirical evidence to show the efficacy of our approach. New experiments below indeed have further strengthened our work and showed its generality


**New Experiment 1: Cutmix [1]**

We perform extensive cross-validation of the **CutMix** hyperparameters on **WideResNet28-10** and **ResNet50** for the **CIFAR10** and **CIFAR100** datasets ($\alpha\in [0.1,0.2,0.3,1.0,10,20]$). Here we report the top performing ones (the cross-validation leverages the in-domain accuracy). Our cross-validation finds, consistently with the CutMix paper, that 1.0 is a good choice for the hyperparameter.

_CIFAR-10 Results_

|  | IND  | DS  | | | OOD   | | |
|-|-|-|--|--|--|--|--|
| | CIFAR10| CIFAR-C| CIFAR10.2| CIFAR10.1| CIFAR100 | SVHN  | Tiny-ImageNet |
|  | Acc     | Acc       | Acc   | Acc   | AUROC | AUROC | AUROC|
| **R50** |||||||
| RegMixup (Ours)| **96.71**   | **81.18**     | **86.72** |  **91.58** | **89.63** | **95.39** | 90.04         |
| CutMix   | 96.51   | 71.09     | 85.25 | 90.50 |  86.90 | 86.62 | **90.30**         |
| **WRN**|||||||
| RegMixup (Ours) | **97.46**   | **83.13**  | **88.05** |  **92.79** | **89.63** | **96.72** | **90.19**         |
| CutMix| 97.15| 73.39| 86.85  | 92.55 | 88.64 | 91.11 | 88.05|

_CIFAR-100 Results_

||IND|DS | OOD   |||
|-|:---:|---:|-|-|--|
| | CIFAR100 | CIFAR-C | CIFAR10 | SVHN  | Tiny-ImageNet |
| **R50**| Acc| Acc| AUROC | AUROC | AUROC|
| RegMixup (Ours) |   **81.52**  |**57.64** | **79.44** | **88.66** | **82.56**|
| CutMix|   81.30  |   48.66 | 78.92 | 81.40 | 80.15|
| **WRN**||||||
| RegMixup (Ours) | **83.25**| **59.44**   | 81.27 | **89.32** | **83.13**|
| CutMix| 83.20    | 51.16| **82.19** | 88.78 | 82.32|

Conclusions: **RegMixup outperforms CutMix** on
- in-distribution accuracy (not very significantly)
- covariate-shift (both synthetic and natural) accuracy (_significant margin_)
- OOD detection (_significant margin_)

**New Experiment 2: ViT-inspired technique that randomly switches between Mixup and CutMix (ViT-Mixup+CutMix) [2]**

To reduce the hyperparameter search space (there are two of them now, _making the hyperparameter search harder with respect to RegMixup_), we fix the $\alpha$ of Mixup to 0.3 and ablate the $\alpha$ for CutMix.

_CIFAR-10 Results_

| | IND| DS | | | OOD   |  | |
|-----|---|---|---|---|--|-|----|
| | CIFAR10| CIFAR-C| CIFAR10.2| CIFAR10.1| CIFAR100 | SVHN  | Tiny-ImageNet |
|| Acc| Acc| Acc   | Acc   | AUROC | AUROC | AUROC|
| **R50** |||||||
| RegMixup (Ours) | 96.71   | **81.18**     | **86.72** |  **91.58** | **89.63** | **95.39** | **90.04**|
| ViT-Mixup+CutMix  | **96.81**  | 75.50  | 86.10 | 91.05  | 89.28 | 95.10 | 89.50|
| **WRN**    | | || | ||
| RegMixup (Ours) | 97.46   | **83.13**  | **88.05** |  **92.79** | **89.63** | **96.72** | **90.19**|
|ViT-Mixup+CutMix | **97.58**   |  76.49   | 87.70   | 92.75  | 85.75 | 92.66 | 86.77|

_CIFAR-100 Results_
|  |IND|DS | OOD|||
|---|:--:|--:|--|-------|--|
| | CIFAR100 | CIFAR-C | CIFAR10 | SVHN  | Tiny-ImageNet |
| **R50**| Acc| Acc| AUROC | AUROC | AUROC|
| RegMixup (Ours)  |   81.52  |   **57.64** | 79.44 | **88.66** | **82.56**|
| ViT-Mixup+CutMix |   **81.98**  |   52.84 | **80.57** | 86.77 | 80.79|
| **WRN**|||| ||
| RegMixup (Ours)  | 83.25    | **59.44**   | **81.27** | 89.32 | **83.13**|
| ViT-Mixup+CutMix | **83.87**    | 54.07   | 79.20 | **89.99** | 81.01|

Conclusions
- ViT-Mixup+CutMix outperforms RegMixup on in-distribution accuracy (**not significantly**)
- Note, ViT-Mixup+CutMix uses **two hyperparameters to cross-validate**
- RegMixup outperforms ViT-Mixup+CutMix on covariate-shift accuracy with **significant** margins
- Except in 2 cases out of 12, RegMixup outperforms ViT-Mixup+CutMix on OOD detection tasks with **significant** margins

**These additional results will be included in our final draft**

[1] CutMix: Regularization Strategy to Train Strong Classifiers with Localizable Features, ICCV19

[2] An Image is Worth 16x16 Words: Transformers for Image Recognition at Scale, ICLR2021

---

### Author Response · Authors · 2022-08-02
**Analyzing the value of $\alpha$ in Mixup | Thorough additional experiments**

Please note that the statement that low $\alpha$ is beneficial for Mixup is in agreement with:
 - the **original Mixup paper** [1]
- a very recent paper performing an **extensive ablation** of the training techniques on ResNets [2].

It is true that in some papers in the literature a $\alpha$ close to 1 is used, here we discuss a couple of them, and _we invite the reviewer to suggest additional relevant papers_ so that we can inspect them further

- in the **CutMix** paper [3] (PyramidNet200) the authors **only consider 0.5 and 1.0** for the cross-validation. We provide thorough cross-validation below.
- in **YOCO** [4] (DenseNet-121 and others) **no cross-validation** is mentioned, we are not sure how $\alpha$ 1.0 is chosen. We provide thorough cross-validation below.
- As for the **Transformer literature**, several augmentation and regularization techniques are combined together, and it is difficult to disentangle the contribution of each component. In any case, as reported in [2], the $\alpha$ for DeiT training is set to 0.8 (hence it is still a bimodal distribution with peaks in 0 and 1, although very close to the uniform). It is still far from employing $\alpha$ = 10 or $\alpha$ = 20 as we do.

**$\alpha$ cross-validation results for Mixup**: We report the cross-validation results below (on the validation splits taken from the training set for C10 and C100, and from the test set for ImageNet) that lead to the **choice of the hyperparameters we used in the paper**, both for CIFAR 10, CIFAR 100 and ImageNet on WideResNet28-10 and ResNet50.

|          | WRN28-10 | WRN28-10 | ResNet50 | ResNet50 |
|----------|----------|----------|----------|----------|
| $\alpha$ | CIFAR10  | CIFAR100 | CIFAR10  | CIFAR100 |
| 0.1      | 96.06    | 81.04    | 95.35    | 79.60    |
| 0.2      | 96.46    | 80.91    | 95.21    | 80.11    |
| 0.3      | **96.77**    | **81.06**    | **95.36**    | **80.31**    |
| 0.4      | 96.71    | 81.01    | 95.26    | 78.93    |
| 0.5      | 96.70    | 80.99    | 95.28    | 78.931   |
| 1        | 96.74    | 80.66    | 94.96    | 78.79    |
| 5        | 96.62    | 79.84    | 94.98    | 77.74    |
| 10       | 96.54    | 79.24    | 94.94    | 75.76    |
| 20       | 96.26    | 78.40    | 95.16    | 75.56    |

Due to the cost of training on ImageNet, we considered a restricted set of hyperparameters for ImageNet as presented below

|          | ResNet50 |
|----------|----------|
| $\alpha$ | ImageNet |
| 0.1      | **77.10**    |
| 0.2      | 77.02    |
| 1        | 76.19    |
| 10       | 72.17    |
| 20       | 71.51    |

**Further cross-validation of $\alpha$ for CutMix and YOCO architectures**: For completeness, we performed a thorough cross-validation of the hyperparameters for two architectures shown in the CutMix [3] (PyramidNet200) and YOCO [4] (DenseNet121) papers on CIFAR-10 and CIFAR-100:

|          | DN-121  | DN-121   | Pyr-200 | Pyr-200  |
|----------|---------|----------|---------|----------|
| $\alpha$ | CIFAR10 | CIFAR100 | CIFAR10 | CIFAR100 |
| 0.1      | 95.89   | 80.54    | 96.71   | 82.34    |
| 0.2      | 96.10   | **80.80**    | 96.70   | 82.17    |
| 0.3      | **96.21**   | **80.80**    | 96.67   | 81.70    |
| 0.4      | 96.06   | 79.71    | 96.79   | **82.62**    |
| 0.5      | 95.98   | 80.17    | **96.92**   | 81.90    |
| 1        | 96.07   | 79.08    | 96.89   | 81.80    |
| 10       | 95.93   | 75.76    | 96.69   | 79.50    |
| 20       | 95.74   | 76.03    | 96.60   | 78.75    |

Conclusions:
- on both C10 and C100 **increasing $\alpha$ produces suboptimal performance** (and reduced with respect to low $\alpha$ ).
- While on C10 the effect of increasing $\alpha$ can be small, on C100 the differences are sharp.
- With respect to [3] we find $\alpha$ =0.5 and $\alpha$ =1.0 to have very similar results on C10 (PyramidNet200), but $\alpha$ =0.4 (_which was not considered in their cross-validation_) to significantly outperform $\alpha$ = 0.5.
- As for **ImageNet** experiments, our training setup (as described in Appendix A.2 and as reproducible using the timm library code) produces optimal value for very low $\alpha$ (in agreement with [1,2]) and decreased performance for high $\alpha$ (in agreement with [1]).

We hope this evidence will convince the reviewer that choosing lower $\alpha$ is beneficial for Mixup and that $\alpha$ >> 1 degrades the performance and is **empirically validated by extensive experiments**. We kindly ask the reviewer to cite specific papers (if different from the ones above) they would like us to closely inspect and discuss further.

[1] mixup: Beyond Empirical Risk Minimization, Zhang et al. ICLR 2018

[2] Resnet strikes back: an improved training procedure in timm, Wightman et al. ImageNet PPF Workshop NeurIPS 2021

[3]: CutMix: Regularization Strategy to Train Strong Classifiers with Localizable Features, ICCV 19.

[4]: You Only Cut Once: Boosting Data Augmentation with a Single Cut, ICML 22.

---

### Meta-Review · Area_Chair_CzhH · 2022-08-30

**Recommendation:** Accept
**Confidence:** Less certain

**Metareview:**

The paper presents a surprisingly simple modification to mixup regularization, which results in a consistent improvement over standard mixup both in the original dataset and in the out-of-distribution setting. All of the reviewers agree that the paper adds to the literature despite its simplicity, and accordingly, I recommend acceptance.

**Award:**

No

---

### Decision · Program_Chairs · 2022-09-14

Accept